# Bottom marine heatwaves along the continental shelves of North America

Dillon J. Amaya [1] ✉, Michael G. Jacox [1,2], Michael A. Alexander [1], James D. Scott [1,3], Clara Deser [4], Antonietta Capotondi [1,3] & Adam S. Phillips [4]

Recently, there has been substantial effort to understand the fundamental characteristics of warm ocean temperature extremes−known as marine heatwaves (MHWs). However, MHW research has primarily focused on the surface signature of these events. While surface MHWs (SMHW) can have dramatic impacts on marine ecosystems, extreme warming along the seafloor can also have significant biological outcomes. In this study, we use a high-resolution (~8 km) ocean reanalysis to broadly assess bottom marine heatwaves (BMHW) along the continental shelves of North America. We find that BMHW intensity and duration varies strongly with bottom depth, with typical intensities ranging from ~0.5 °C–3 °C. Further, BMHWs can be more intense and persist longer than SMHWs. While BMHWs and SMHWs often co-occur, BMHWs can also exist without a SMHW. Deeper regions in which the mixed layer does not typically reach the seafloor exhibit less synchronicity between BMHWs and SMHWs.

Warm ocean temperature extremes−known as marine heatwaves (MHW)−can dramatically impact the overall health of marine ecosystems around the globe, including changing the regional distribution of marine species, altering primary productivity, and increasing the risk of negative human-wildlife interactions[1–6]. As a result, there has been a considerable effort to characterize the timing, intensity, duration, and physical drivers of both individual and composite MHW events[2,7–16]. However, MHW research has primarily focused on sea surface temperature (SST) extremes. Such constraints are often convenient since SST is a useful predictor of certain species distributions[17–19] and can map onto shifts in the many physical and biogeochemical ocean variables important to the health of sensitive marine ecosystems[19]. In addition, there are simply many more high-quality observations of the surface ocean than of the subsurface, thus making analysis of SST extremes and their impacts more straightforward.

There have been some efforts in recent years to describe MHWs throughout the water column using limited subsurface data. For example, some studies have used temperature and salinity data from Argo profiles to investigate MHW events in the Northeast Pacific and Tasman Sea[20,21]. Still others have used moored buoys, underwater gliders, and aggregated gridded observations to analyze the subsurface characteristics of MHW events off the east Australian coast[22], the north Australian coast[23], the Mediterranean Sea[24], the Gulf of Mexico[25], and the Northwest Atlantic continental shelf[26].

Despite these recent advances in understanding the subsurface nature of MHWs, there have been no targeted efforts to characterize temperature extremes on the ocean bottom along continental shelves (hereafter referred to as bottom marine heatwaves; BMHW). Intense bottom temperature changes can have unique and dramatic impacts on the productivity and organization of demersal species found along the continental shelf. For example, bottom water temperature anomalies (BWTA) have been linked to declines in Gulf of Alaska Pacific cod abundance[27], shifts in the occurrence of demersal fish in the California Current System[28], the redistribution of invasive lionfish

[1]Physical Sciences Laboratory, Earth System Research Laboratory, National Oceanic and Atmospheric Administration, 325 Broadway, Boulder, CO 80305, USA. [2]Environmental Research Division, Southwest Fisheries Science Center, National Oceanic and Atmospheric Administration, 99 Pacific St #255A, Monterey, CA 93940, USA. [3]Cooperative Institute for Research in Environmental Sciences, University of Colorado Boulder, 216 UCB, University of Colorado Boulder campus, Boulder, CO 80309, USA. [4]National Center for Atmospheric Research, 1850 Table Mesa Dr, Boulder, CO 80305, USA. ✉e-mail: dillon.amaya@noaa.gov

along the Southeast US (SEUS) continental shelf[29], altered recruitment dynamics of Atlantic cod[30], mediation of predation pressure on scallops throughout the Northeast US (NEUS) continental shelf[31], and shifts in disease onset patterns in lobster[32].

In addition to the unique biological impacts of bottom temperature variability, it is unclear whether surface MHW (SMHW) events as measured by sea surface temperature anomalies (SSTA) are a suitable physical proxy for BMHW conditions. For example, while M. Alexander, J. Scott, D. Amaya, C. Deser, M. Jacox, A. Phillips (in preparation) (hereafter referred to as Alexander et al. in preparation) show that BWTAs and SSTAs can be highly correlated in shallow coastal regions, they also suggest that the interaction of subsurface ocean currents with complicated bathymetric features may lead to BWTAs that not only peak at different depth ranges, but are also more intense than the corresponding SSTAs. Additionally, since bottom waters well below the mixed layer do not lose heat to the atmosphere, these BWTAs (and possibly BMHWs) may persist longer than SSTAs at the same location (Alexander et al. in preparation). Given the clear relevance of bottom temperature variability to the health of marine ecosystems as well as the varying physical evolution of BWTAs relative to the surface, it is essential to assess the intensity, duration, and spatiotemporal structure of BMHWs as unique events.

In this study, we use a state-of-the-art high-resolution (1/12˚ or ~8 km) ocean reanalysis to generate a large-scale assessment of BMHW statistics throughout North American Large Marine Ecosystems (LMEs). We find that BMHW intensity and duration variesy strongly with ocean bottom depth. Additionally, not only do BMHWs tend to persist longer than their surface counterparts, but there are many regions where BMHW intensity tends to exceed SMHW intensity for the same location. Finally, we show that BMHW and SMHW events can co-occur, particularly in shallower regions where deep mixed layers link surface and bottom waters. However, we also find that some BMHWs are generated without a clear surface signature, highlighting the importance of maintaining routine subsurface ocean monitoring systems.

## Results

### Bottom marine heatwave intensity, duration, and spatial extent

Within each LME, the average intensity of BMHW events can strongly vary in space, with typical BMHW anomalies ranging from as low as 0.5 °C for deeper portions of the continental shelves in all LMEs (see Fig. 1 for the bathymetric details of each region) to as high as 5 °C for large portions of the Gulf of California at ~100 m depth (Fig. 2). In LMEs with more highly varying bathymetry along the continental shelf (e.g., Gulf of Alaska, NEUS, Scotian, and Labrador LMEs), the spatial distribution of average BMHW intensity is complex, with more intense events often linked to distinct bathymetric features (compare panels of Fig. 2 with those in Fig. 1). For example, in the NEUS LME, the intensity of BMHW events clearly differs between the relatively shallow Mid-Atlantic Bight (MAB; the coastal region between ~35˚N–41˚N) and deeper Gulf of Maine (NEUS LME basin north of 41˚N; Figs. 1g and 2g). Additionally, the southwest portion of the Scotian continental shelf is characterized by deep basins (bottom depths > 200 m) that are punctuated by shallow banks (bottom depths <100 m). As a result, average BMHW intensity varies strongly in this region, with more intense events typically found along the shallow banks and weaker events in the deeper basins (Figs. 1h and 2h). These bathymetrically diverse LMEs stand in contrast to those with relatively narrow/sharp continental shelves (e.g., California Current and Gulf of California) or relatively wide/smooth continental shelves (e.g., East Bering Sea, Gulf of Mexico, and SEUS). In these less bathymetrically varying LMEs, the distribution of average BMHW intensity is more uniform in space and tends to peak in regions close to the coast with shallower bottom depths (Figs. 1 and 2).

In an effort to quantify how BMHW intensity varies with ocean bottom depth, we present scatter plots of BMHW average intensity versus the corresponding ocean bottom depth at each grid cell for each LME (Fig. S1). Next, we convert these scatter plots into two-dimensional probability histograms for each LME by binning the grid cells into various intensity and depth intervals (Fig. 3). There is a clear negative relationship between BMHW intensity and ocean bottom depth in the East Bering Sea and Gulf of Alaska LMEs (Fig. 3a, b). The Spearman correlation coefficients between the intensity and ocean bottom depth across all grid cells within each of these LMEs are $R = -0.84$ and $-0.9$, respectively. In other regions, however, the relationship between BMHW average intensity and depth is more complicated. For example, in the Gulf of California LME and for portions of the California Current LME (Fig. 3c, d), BMHW intensity and bottom depth are positively correlated from the surface to ~100 m and then negatively correlated from ~100–200 m before asymptotically leveling off from 200–400 m. As a result, the warmest BMHW intensities occur at intermediate bottom depths of 50–100 m, with average values in this depth interval of 2.9 °C and 3 °C for the California Current and Gulf of California, respectively (e.g., Fig. 3c, d, gray dots). In the California Current LME, this non-linear BMHW-depth relationship is only found in the southern portion of the domain (Fig. S1c), while BMHW intensities tend to decrease roughly linearly with depth in the northern portion of the LME ($R = -0.95$), similar to those in the East Bering Sea and Gulf of Alaska LMEs.

In contrast, the LMEs found along the Gulf of Mexico and the North American east coast do not show as clear of a BMHW-depth relationship (Fig. 3e–i), which may be due to the complicated bathymetric and oceanographic features in these LMEs. For example, bottom temperature variability in the MAB is likely dominated by instabilities in the shelf-break front[33], while bottom temperature variability in the Gulf of Maine is associated with strong tidal mixing[34] and possibly advection by the Labrador Current[35] or the mixing of Gulf Stream waters at depth via the Northeast Channel[36]. These different flow-bathymetry interactions may explain the BMHW intensity differences discussed previously (e.g., Fig. 2g) and ultimately lead to a scattered BMHW-depth relationship in which average intensities peak at ~100 m, but otherwise do not closely scale with bottom depth (Fig. 3g).

The average duration of BMHW events also exhibit strong spatial variations across the different LMEs (Fig. 4). In several LMEs, longer duration BMHWs are associated with deeper portions of the continental shelf, such as in western portion of the East Bering Sea, the Gulf of Maine in the NEUS, the southeastern portion of the Scotian Shelf, and the northeastern portion of the Labrador LME (comparing Figs. 1a, g, i with Figs. 4a, g, i). Outside of these regions, however, the BMHW duration is highly variable and not clearly linked to specific bathymetric features. The California Current LME shows a clearer pattern of average duration, with BMHWs in the southern portion of the domain lasting longer than BMHWs in the northern portion of the domain (Fig. 4c). Similarly, the BMHW events in the Gulf of Mexico and SEUS LMEs show generally uniform average durations of ~1.5 months (Fig. 4e, f). The majority of the LMEs exhibit noisy patterns of BMHW longevity that are not strongly related to bottom depth (Fig. S2).

In order to diagnose the prevalence of BMHWs on the broader LME-scale, we assess the spatial extent of these events with time (Fig. 5). In the East Bering Sea, there are two major periods of widespread and prolonged BMHW events, including several from 2002–2006 and again from 2015–2018 (Fig. 5a). At its peak, the BMHW event beginning in 2016 encompasses 60% of the total area of the LME with a maximum average intensity of 2.5 °C. These prolonged and widespread BMHW conditions correspond with a well-known warming event which produced myriad marine ecosystem impacts[37].

The Gulf of Alaska, California Current, and Gulf of California LMEs similarly feature two pronounced periods of prolonged and spatially

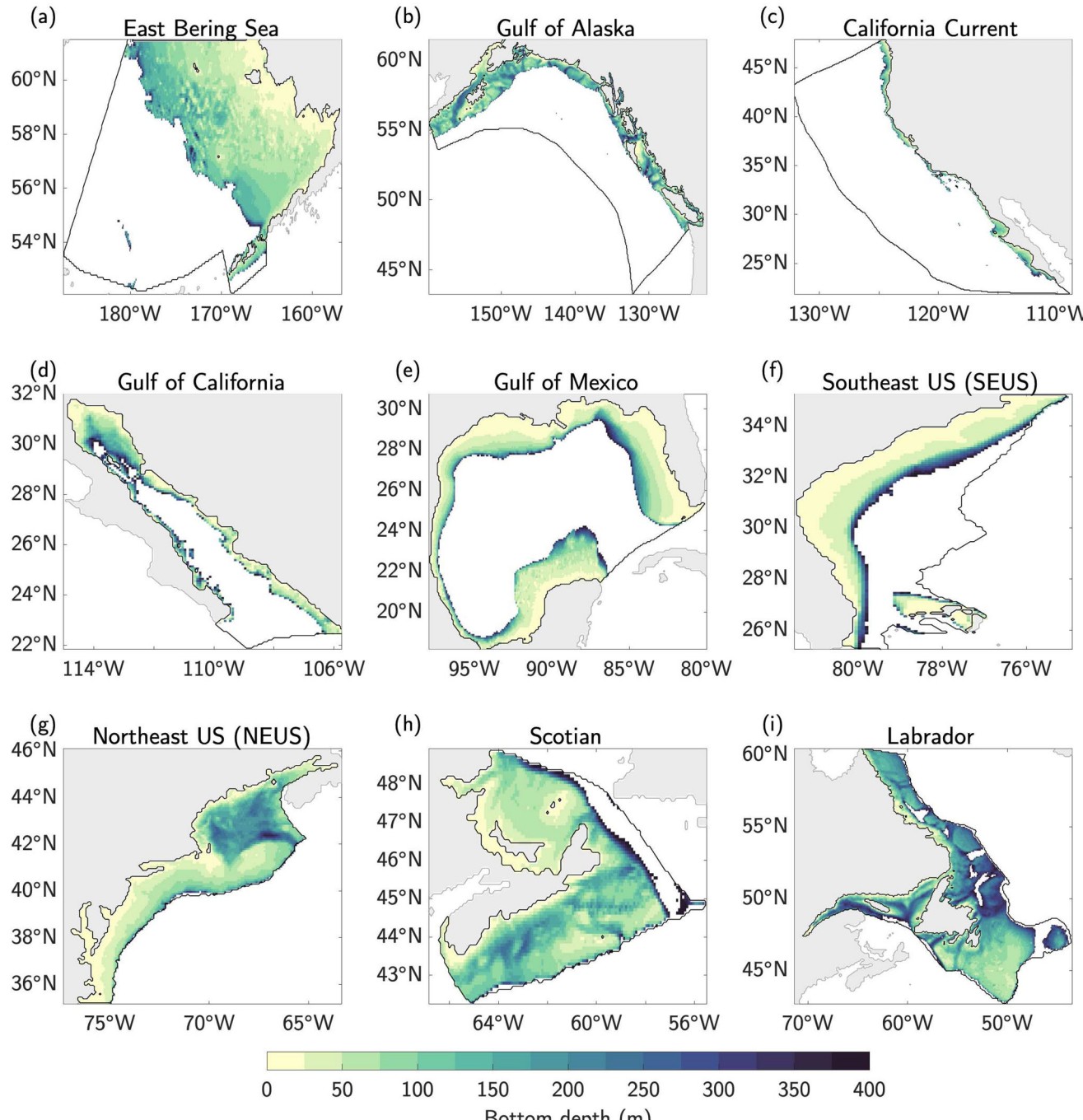

**Fig. 1 | Bathymetry along the continental shelves of North America. a–i** Ocean bottom depth (m) for each of the nine Large Marine Ecosystems (LMEs) along North American coastlines. Ocean grid cells with bottom depths deeper than 400 m are shaded white. Land surfaces are shaded gray. Black contours outline each LME.

widespread BMHW events, including 1997-1998 and 2014–2017 (Fig. 5b, d). The BMHW in 1997/1998 is the most widespread and intense of the two time periods, with peak areal extents of 0.72, 0.96, and 0.87 and peak average intensities of 1.6 °C, 3.5 °C, and 5 °C for the Gulf of Alaska, California Current, and Gulf of California LMEs, respectively. These intense BMHW conditions correspond with the 1997/1998 El Niño event, suggesting that BMHW events along the North American west coast, like their surface counterparts, may be linked by large-scale climate forcing related to the El Niño-Southern Oscillation (ENSO). We will return to this point in the Discussion section.

The BMHW events in the Gulf of California, California Current, and Gulf of Alaska LMEs from 2014–2016 feature two peaks, one in 2014 to

early 2015 and another in late 2015 to 2016. The first of these peaks correspond with the evolution of a series of major Northeast Pacific MHWs[12,14]. During this time period, large-scale atmospheric circulation anomalies produced strong surface winds along the North American west coast, contributing to the development of MHW conditions in the California Current System[12,38,39]. Surface forcing associated with these wind anomalies likely contributed to the intensity and areal extent of the 2014-2015 BMHW conditions seen in the California Current LME, which peaked at 67% LME coverage and an average intensity of 2.6 °C. The intensity, spatial extent, and persistence of widespread BMHW conditions throughout the Gulf of California, California Current, and Gulf of Alaska from late 2015 into 2016 consistent with a series of downwelling coastally trapped waves that propagated up the North

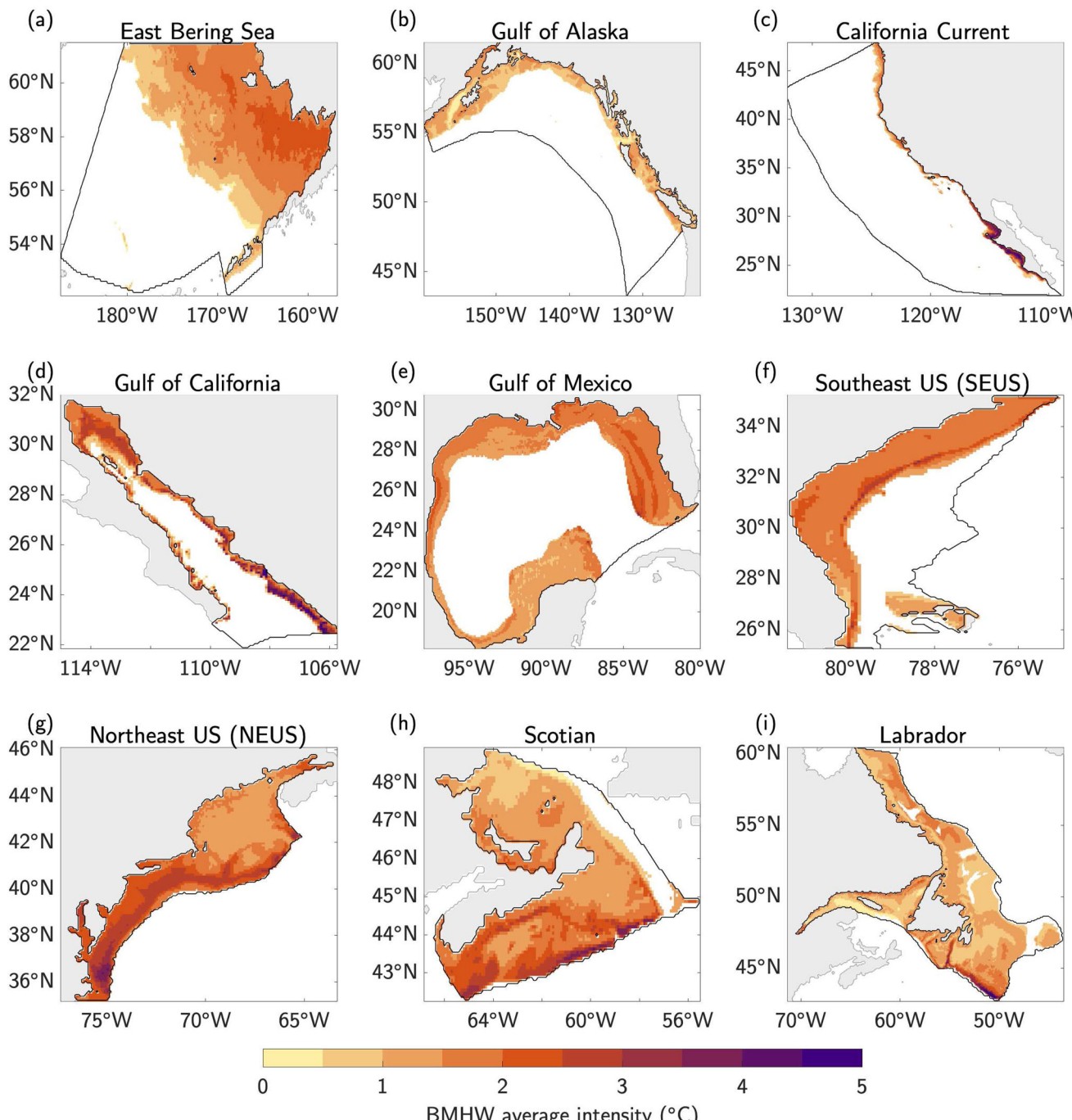

**Fig. 2 | Average intensity of bottom marine heatwaves. a–i** Bottom water temperature anomalies (°C) averaged during all bottom marine heatwave (BMHW) months from 1993–2019 in each Large Marine Ecosystem (LME).

American coastline during the development of the 2015/2016 El Niño[40]. In all, BMHW conditions from 2015-2016 peaked at 0.81, 0.64, and 0.61 areal coverage and a peak average intensity of 1.4 °C, 3.0 °C, and 3.2 °C. for the Gulf of Alaska, California Current, and Gulf of California LMEs, respectively.

The BMHW conditions in the Gulf of Mexico and SEUS are generally less coherent (e.g., more instances with fractional area <0.5) on the LME-scale (Fig. 5e, f) than those along the Pacific coast, suggesting the presence of within-LME subregions of isolated BMHW activity that may be tied to specific bathymetric features or local processes. Although the spatial extent of BMHW conditions in the NEUS, Scotian Shelf, and Labrador LMEs are generally lower than those along the Pacific coast, these LMEs do show prolonged BMHW events during

1999-2000 and 2011-2012. In particular, the BMHW in 2011-2012 peaked at 0.64, 0.5, and 0.57 areal coverage with peak average intensities of 3 °C, 3.8 °C, and 1.8 °C for the NEUS, Scotian Shelf, and Labrador LMEs, respectively. The coherent BMHW conditions in 2011-2012 may be related to large-scale atmospheric forcing associated with the broader 2012 Northwest Atlantic MHW[41,42].

**Comparing bottom and surface marine heatwaves**

In the East Bering Sea, the Gulf of Alaska, and the northern portions of the California Current LME, SMHW intensity tends to be ~0.5 °C–1 °C warmer than BMHW intensity (Fig. 6a, c; blue shading), while in the southern California Current, Gulf of California, Gulf of Mexico, and SEUS LMEs, an average BMHW is anywhere from

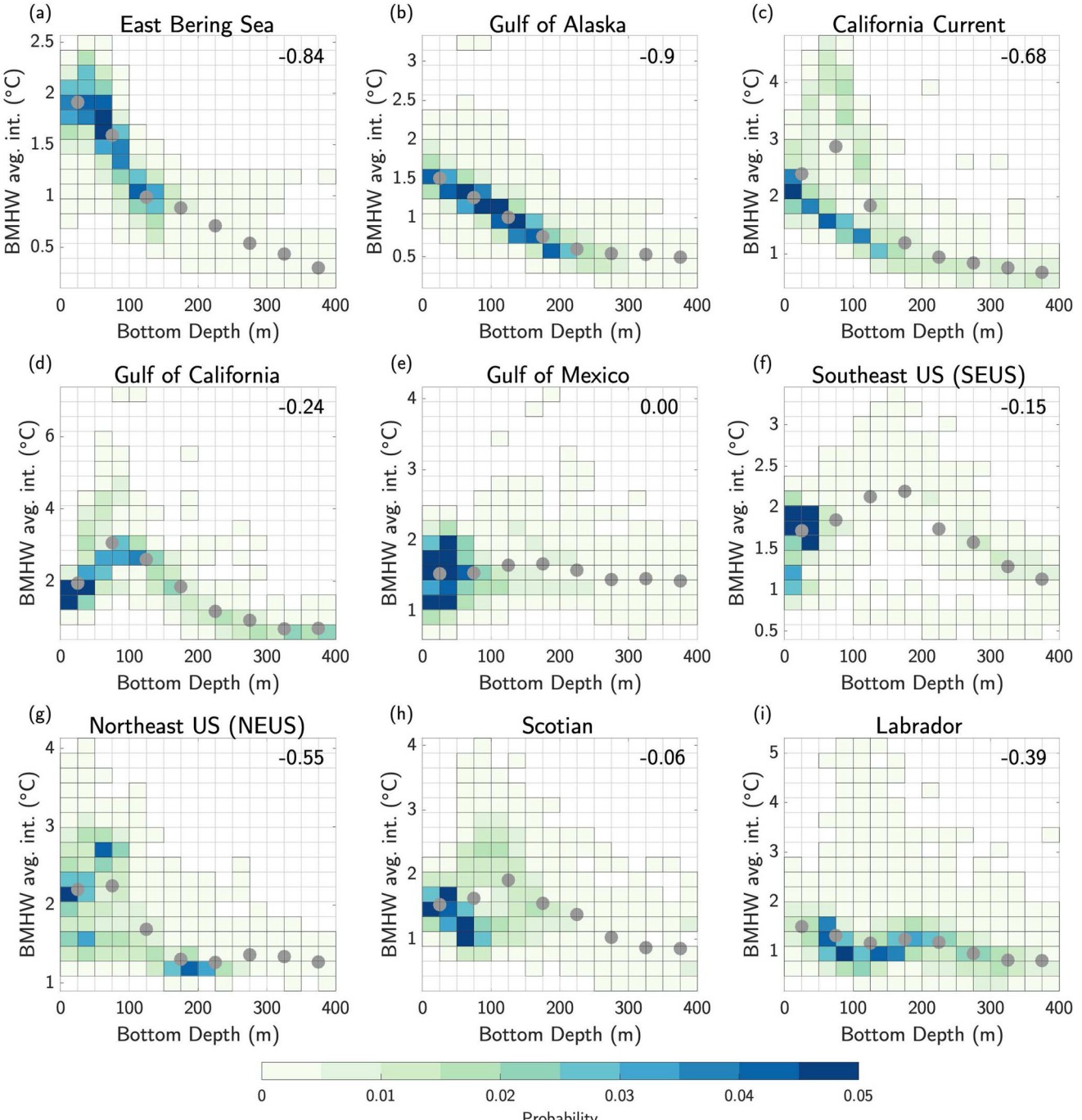

**Fig. 3 | Bottom marine heatwave intensity variations with ocean depth. a–i** Two-dimensional histograms of bottom marine heatwave (BMHW) average intensity (°C) versus ocean bottom depth (m) in each Large Marine Ecosystem (LME). Shading indicates the probability that a grid cell (with bottom depth <400 m) falls within a given intensity-depth interval. The Spearman correlation between BMHW average intensity and bottom depth across all grid cells within each LME is shown in the top right of each panel. Gray dots indicate the BMHW average intensity averaged across regular depth intervals of 50 m. The position of each dot along the x-axis represents the center of the depth interval used for averaging. For example, the first gray dot is positioned at 25 m, and represents the BMHW average intensity averaged across all grid cells with bottom depths of 0–50 m.

0.5 °C–2.5 °C warmer than an average SMHW (Fig. 6c–f; orange/red shading). The NEUS, Scotian, and Labrador LMEs show more varied intensity difference patterns that are likely driven by complex current-bathymetry interactions. For example, there are once again clear differences between the MAB and the Gulf of Maine in the NEUS LME. In the MAB, an average BMHW is ~1 °C–2 °C warmer than an average SMHW, but SMHWs tend to be ~0.5 °C warmer than BMHWs in the interior portions of the Gulf of Maine where the seafloor is deepest (Fig. 6g).

Compared to average BMHW intensity, average SMHW intensity is less variable in space within a given LME (Fig. S3). As expected, the difference between average BMHW and SMHW intensity in each LME is close to zero for the shallowest ocean bottom depths, where SMHWs and BMHWs converge (Fig. S4). In each LME, the average BMHW persists longer than the average SMHW at almost every location (Fig. 7), likely due to the fact that SMHWs are damped by the atmosphere via turbulent heat fluxes while the ocean bottom is largely insulated from such changes. This hypothesis is supported by the slower

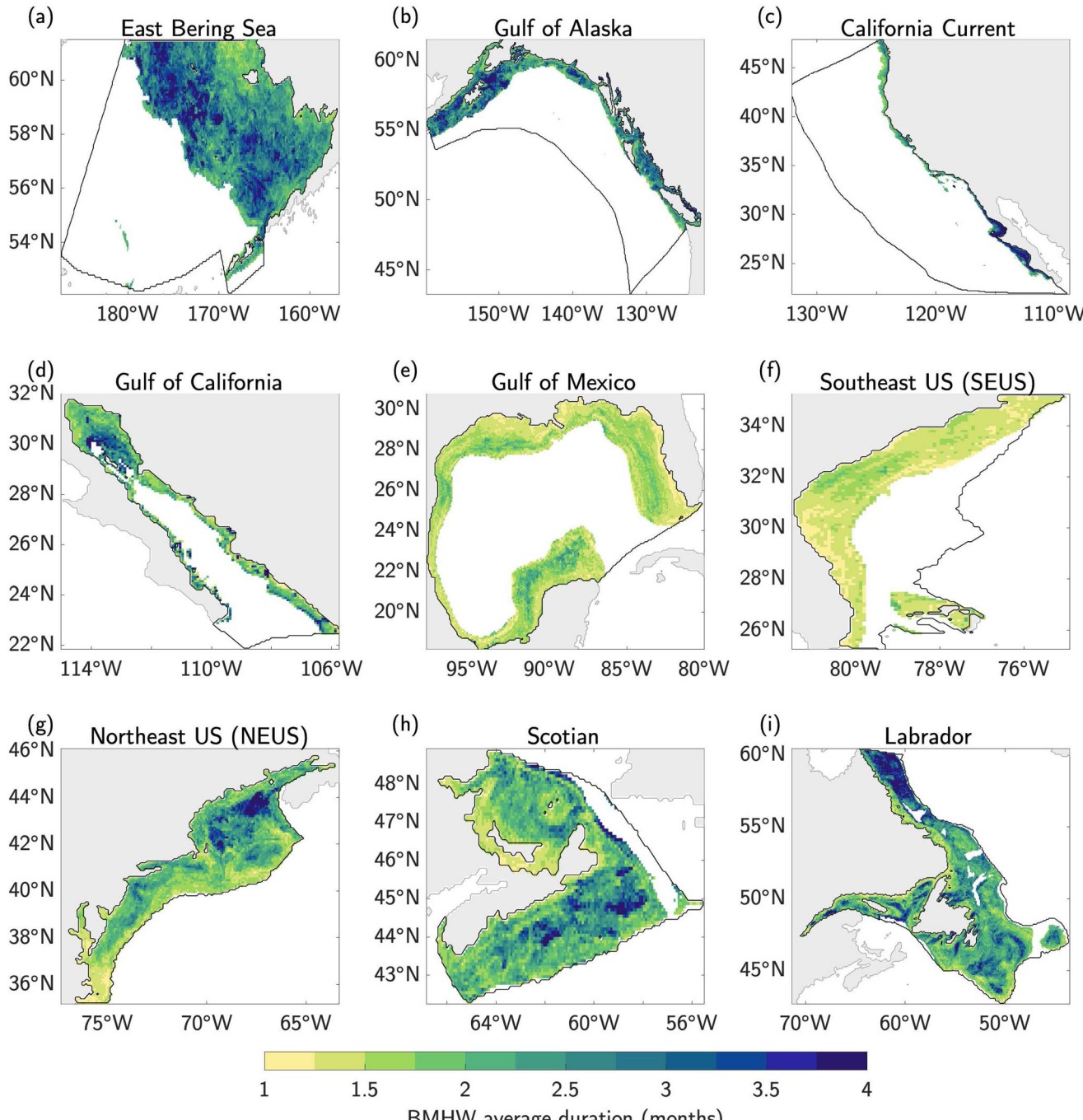

**Fig. 4 | Average duration of bottom marine heatwaves. a–i** The average duration (months) of all bottom marine heatwave (BMHW) events from 1993–2019 in each Large Marine Ecosystem (LME).

decorrelation timescale[43] of BWTAs compared to SSTAs in each region (Figs. S5–S7), leading to a slower breakdown of BMHW conditions from one month to the next.

Spatial coherence tends to be higher for SMHWs than for BMHWs, as indicated by the total number of months in which SMHW or BMHW conditions are widespread (i.e., exceeding 50% of an LME's area; Fig. 8, numbers next to LME name). In every LME, there are more months with widespread SMHW conditions than with widespread BMHW conditions. Despite these differences, however, the BWTAs associated with spatially coherent BMHWs are often warmer than the concurrent overlying SSTAs (Fig. S8). Of note are the anomalous temperature differences between the ocean bottom and surface during the 1997/1998 and 2015/2016 North Pacific MHW events. In the California

Current LME, BWTAs were greater than 1 °C warmer than the co-located SSTAs during the 1997/1998 event, while BWTAs in the Gulf of California LME were greater than 2.5 °C warmer than the surface anomalies during the 1997/1998 El Niño and the 2015/2016 MHW (Fig. S8c, d).

In each region, there are periods of spatially widespread SMHW events with peak areal extents of greater than 0.5 that closely correspond to concurrent widespread BMHW conditions (Fig. 8). For example, in the East Bering Sea, there are widespread SMHW conditions in 2001, 2003-2004, and 2016 that occur in conjunction with widespread BMHW conditions. The same is true for the Gulf of Alaska, California Current and Gulf of California LMEs from 1997/1998 and again from 2015/2016. There are also periods of widespread SMHW

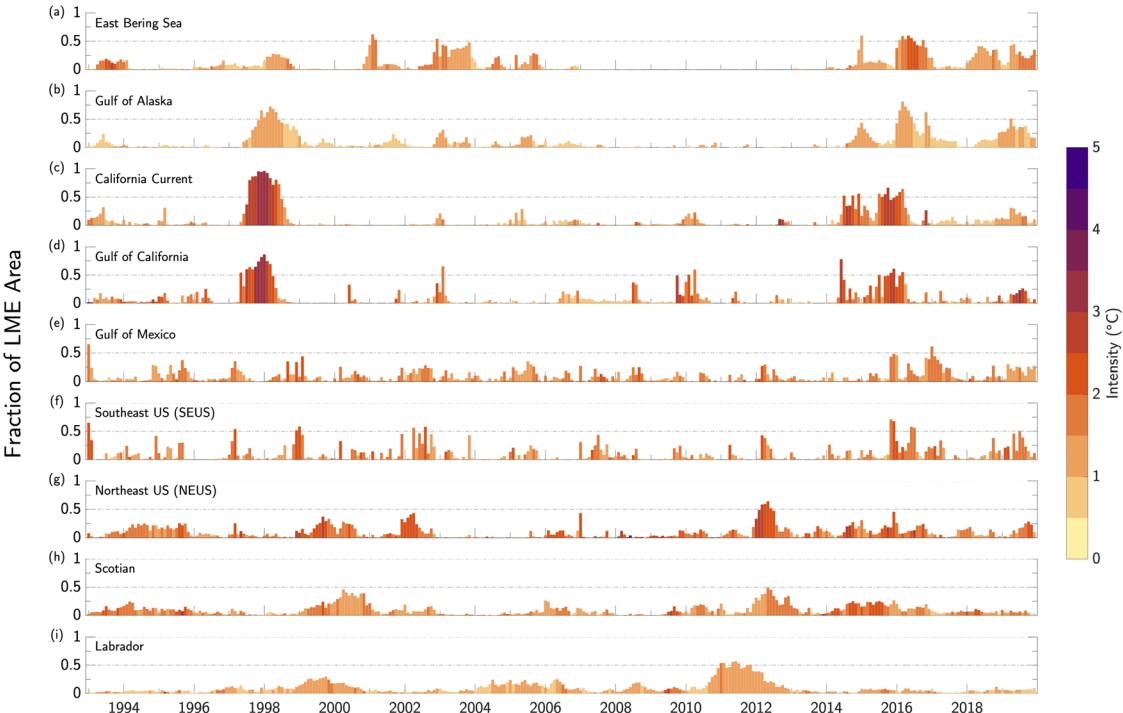

**Fig. 5 | Spatial extent of bottom marine heatwaves. a–i** The fraction of each Large Marine Ecosystem's (LME) area experiencing bottom marine heatwave (BMHW) conditions for each month from 1993–2019. Shading denotes average BMHW intensity (°C) in a given month, as measured by bottom water temperature anomalies averaged across all grid cells experiencing BMHW conditions. Horizontal gray lines mark areal extents of 0.5 and 1. Note only grid cells with bottom depths <400 m were used for areal percentage and intensity calculations.

and BMHW conditions in the Gulf of Mexico and SEUS in 2016/2017 and in the NEUS, Scotian, and Labrador LMEs during the well-known 2012 warming event.

There are, however, some notable differences in the spatial extent of SMHW and BMHW events. In particular, in the Gulf of Alaska and California Current LMEs, while SMHW and BMHW conditions do occur simultaneously during the development of the 1997/1998 El Niño, there is a notable lag between the most widespread (>-0.5 areal extent) SMHW conditions and the most widespread BMHW conditions (Fig. 8b, c). Specifically, a BMHW with large spatial extent follows a widespread SMHW several months later. In the Gulf of Alaska, these BMHW conditions persist as many as 7 months after SMHW conditions have subsided. A somewhat similar lagged relationship is seen in the Gulf of Alaska during the evolution of the weak 2014/2015 El Niño. The differences in timing may be related to the different physical mechanisms relevant to the formation of widespread SMHW or BMHW conditions. For example, the coastal surface ocean in the Gulf of Alaska and California Current LMEs is strongly influenced by large-scale atmospheric teleconnections associated with the Pacific-North American (PNA) pattern[44], which develops rapidly in response to tropical heating associated with ENSO[45]. Whereas the ocean bottom is likely more sensitive to slower adjustments in subsurface currents, the propagation of coastally trapped waves, or vertical displacements of the thermocline[46]. These different mechanisms will be discussed in more detail in the Discussion section. In addition to the differences seen along the North American west coast, there are key differences in the spatial extent of SMHW and BMHW events along the east coast. In the Scotian and Labrador LMEs, a widespread BMHW event begins in 1999 and persists into 2000 (Fig. 8h, i). However, the corresponding surface warming is comparatively disjointed in time. Also in the Labrador LME, there is a persistent and widespread BMHW event from 2010-2013; however, from late 2011 to early 2012 there is not a corresponding SMHW. Instead, surface warming is broken up into two separate events, one in 2011 and another in the winter of 2012/2013.

## Bottom and surface marine heatwave synchrony

In almost every LME, there is a clear pattern of synchrony between BMHWs and SMHWs, with the two co-occurring more often over shallower portions of the shelf (Fig. 9). As a result, for the majority of the LMEs, the synchrony of BMHWs and SMHWs scales with depth following a relationship that can be approximated using a 2nd-order polynomial model fit (Fig. S9). Therefore, the LMEs with wider continental shelves and more area at shallower depths (e.g., East Bering Sea, Gulf of Mexico, and SEUS; Fig. 1) have a larger fraction of their area in which BMHWs and SMHWs co-occur.

The higher synchrony of BMHW and SMHW events for shallow ocean bottom depths may be expected since it is more likely in shallow regions for the mixed layer depth (MLD) to extend to the ocean floor, at which point the physical characteristics (e.g., temperature) of bottom waters would match those near the surface. We test this hypothesis by first calculating the ratio of the time varying MLD to the ocean bottom depth at each grid cell within each LME. We then composite this MLD/bathymetry ratio during months when a BMHW and SMHW are co-occurring (Fig. S10). In each LME, the synchrony of BMHWs and SMHWs is highly correlated with MLD/bathymetry ratio at each grid cell (Fig. 10), such that BMHWs and SMHWs tend to co-occur more frequently when the MLD/bathymetry ratio is high, i.e., the closer to the ocean bottom the MLD reaches during a SMHW, the more likely a BMHW is to occur.

## Discussion

In this study, we analyzed the statistical characteristics of MHW events occurring along the ocean bottom of continental shelves in nine North American LMEs. Using a state-of-the-art high-resolution (1/12° or ~8 km) ocean reanalysis spanning 1993–2019, we found that average BMHW intensity varies strongly across the different LMEs, ranging from as low as 0.5 °C at deeper bottom depths (typically>200 m) in all LMEs to as high as 5 °C for large portions of the Gulf of California at ~100 m depth (Fig. 2). Average BMHW intensity scales roughly linearly

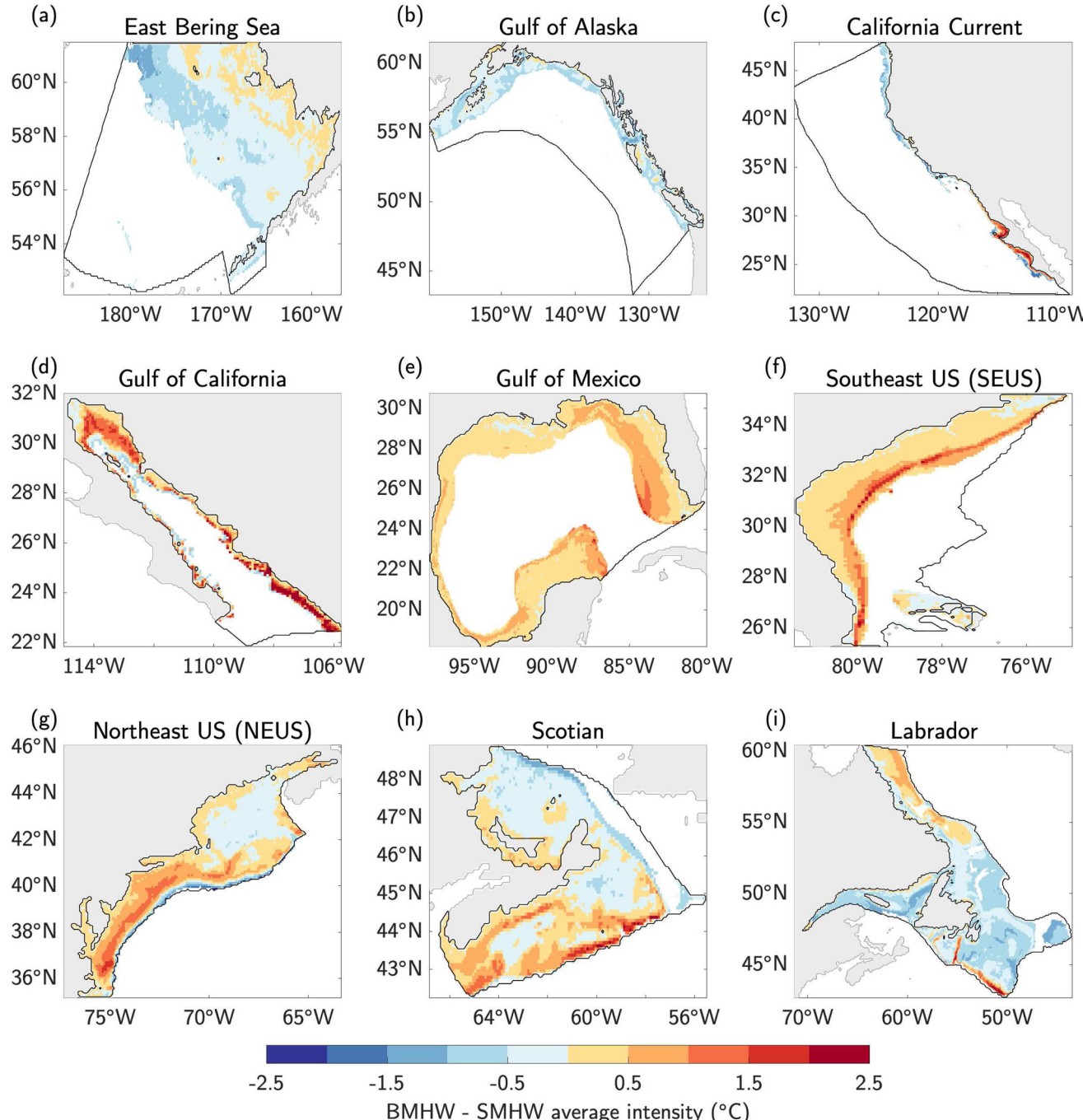

**Fig. 6 | Bottom vs. surface marine heatwave intensity. a–i** Difference between bottom water temperature anomalies (°C) averaged during all bottom marine heatwave (BMHW) months and sea surface temperature anomalies (°C) averaged during all surface marine heatwave (SMHW) months from 1993–2019 in each Large Marine Ecosystem (LME).

with depth in the East Bering Sea and the Gulf of Alaska, while the intensity-depth relationship is more complicated in the other LMEs (Fig. 3). For example, the BMHW intensity-depth relationship is non-linear in the Gulf of California and California Current (although mainly in the southern portion of the domain; Fig. S1c), with peak intensities occurring at -100 m. In the Gulf of Mexico and the North American east coast LMEs, intensity does not clearly scale with depth. Similar to intensity, the average BMHW duration showed regional differences that were related to various physical regimes found within the different LMEs (Fig. 4). For example, BMHW events were longer lived for the Gulf of Maine region of the NEUS LME and the southern portion of the California Current LME. Additionally, we identified periods of spatially

coherent BMHW events within each LME over the past 30 years (Fig. 5), including during the 1997/1998 El Niño in the Gulf of California, California Current, and the Gulf of Alaska and during the well-known 2012 Northwest Atlantic MHW in the NEUS, Scotian, and Labrador LMEs.

We further compared the characteristics of BMHW and surface MHW (SMHW) events and found that not only do BMHW events tend to persist longer than SMHW events nearly everywhere, BMHW intensity exceeds SMHW intensity in many regions (Figs. 6, 7, Fig. S8). We found that widespread BMHW conditions often correspond to similar SMHW conditions within a given LME, but there are also periods of considerable BMHW activity without a clear SMHW signal, such as in the Labrador LME from mid-2011 to early-2012 (Fig. 8). Finally, we

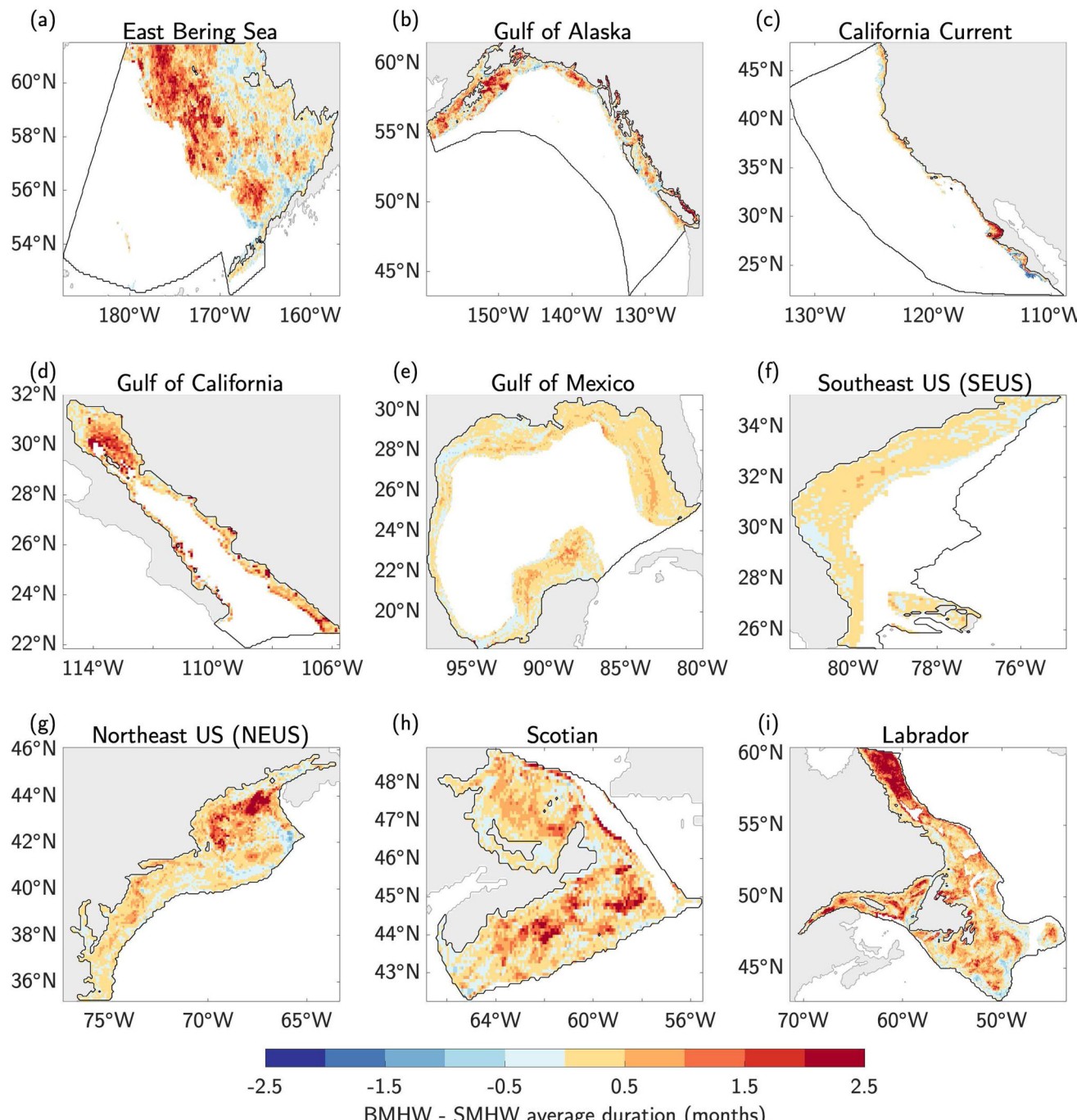

**Fig. 7 | Bottom vs. surface marine heatwave duration. a–i** The difference in average duration (months) between bottom marine heatwaves (BMHWs) and surface marine heatwaves (SMHWs) from 1993–2019 in each Large Marine Ecosystem (LME).

showed that BMHWs and SMHWs are more synchronous in shallow coastal regions where the mixed layer is more likely to reach the ocean floor, thereby linking surface and bottom waters (Figs. 9, 10).

Our results have several important implications for future research into the coastal ocean regions around North America. First, the spatial variations in BMHW intensity and duration and their different relationships with bottom depth suggest that BMHW characteristics are partially governed by flow interactions with prominent bathymetric features, which may differ from one LME to another or even within a given LME (Figs. 1–3 and Figs. S1, S2). For example, there are clear differences in BMHW statistics between the northern and southern portions of the California Current LME and between the Gulf of Maine and MAB regions of the NEUS LME. Additionally, the stronger

spatial variation in BMHW average intensity compared to SMHW average intensity (comparing Fig. 2 and S3) is a further indication that bottom topography is critical in shaping the patterns described in this study. As a result, each LME would benefit from a dedicated focus aimed at diagnosing the unique physical drivers of BMHW events and how these mechanisms relate to different bathymetric features in the respective regions.

Our study does suggest a number of physical mechanisms with which to build testable hypotheses going forward. For example, we noted a likely influence of ENSO in modulating BMHW conditions along the North American west coast (Fig. 5b, d). The time-lag in the peak BMHW spatial extent going from the Gulf of California to the Gulf of Alaska from 1997-1998 and from 2015-2016 is consistent with the

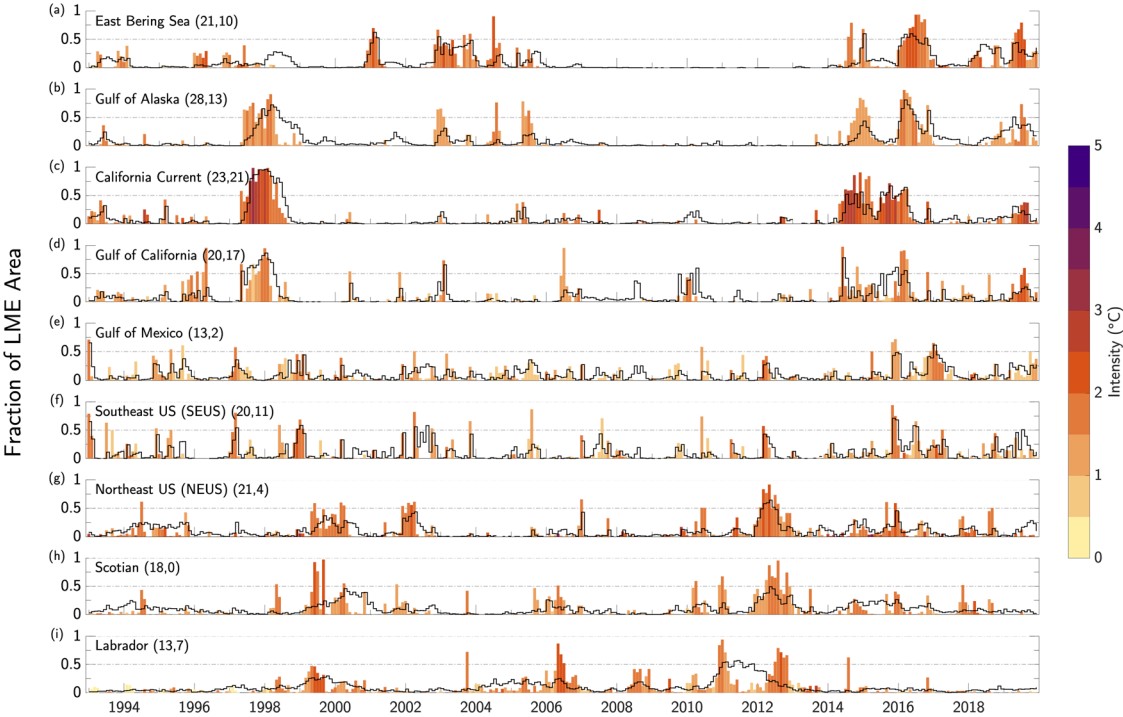

**Fig. 8 | Bottom vs. surface marine heatwave spatial extent. a–i** The fraction of each Large Marine Ecosystem's (LME) area experiencing surface marine heatwave (SMHW) conditions for each month from 1993–2019. Shading denotes average SMHW intensity (°C) in a given month, as measured by sea surface temperature anomalies (SSTAs) averaged across all grid cells experiencing SMHW conditions. Black contours mark fraction of LME's area in bottom marine heatwave (BMHW) conditions (i.e., bar height in Fig. 5). Horizontal gray lines mark areal extents of 0.5 and 1. Note only grid cells with bottom depths <400 m were used for areal percentage and intensity calculations. Numbers next to LME names indicate total number of months with areal extent greater than 0.5 for SMHWs and BMHWs, respectively.

propagation of coastally trapped waves associated with the development of extreme El Niño events in each of these winters. In particular, downwelling coastally trapped waves depress the thermocline along the continental shelf, bathing the shallow near-shore ocean floor in warmer waters from higher in the water column. Several intense downwelling coastally trapped waves were observed during the development of the 1997/1998 El Niño[40], a period when ~5 °C BWTAs were seen in the Gulf of California and California Current System. Given the strong vertical temperature gradients associated with the thermocline, relatively small vertical displacements in its depth can drive BMHWs in all LMEs, not just along the US west coast. Along the North American east coast, we showed that widespread BMHW conditions in the Labrador LME in 2011-2012 preceded widespread BMHW conditions in the Scotian and NEUS LMEs by several months, consistent with advection of anomalies by the Labrador Current. Similarly, Chen et al.[26] showed that BWTAs in portions of the NEUS LME were highly correlated with anomalies upstream (i.e., to the north). We plan to investigate these mechanisms in more detail in future studies.

Our results also provide critical insight into the relationship between SMHW and BMHW events in the near coastal environment. Despite SMHWs having shorter average duration than BMHWs, there have been more instances of high spatial coverage (e.g., >0.5 areal extent) among SMHWs than among BMHWs (Fig. 8). Two likely sources of this discrepancy are: (1) BMHW events are more likely to be geographically constrained by rigid bathymetric features, which would limit the areal extent of BMHWs within a given LME; and (2) Surface wind and cloud radiative forcing are well-known drivers of SMHWs in much of the world's oceans[8,13,47] and surface heat fluxes associated with these atmospheric changes are more likely to have a large horizontal footprint compared to some of the possible drivers of BMHW events (e.g., subsurface currents).

Finally, our results show that BMHW and SMHW events often co-occur in shallow regions very near to coasts (e.g., high synchrony). A similar observation was made by Schaeffer and Roughan[22], who used in situ temperature measurements on the eastern Australian continental shelf to show that SMHW events in this region extend deep into the water column (even reaching the bottom) when upper ocean stratification is low. Our analysis of the relationship between MLD, bottom depth, and SMHW/BMHW synchrony significantly builds on these early ideas by testing this hypothesis more broadly and across more varied physical environments, finding it to be generally true. Additionally, we further demonstrate the possibility of BMHW events with intensities and durations that exceed those of more traditional SST-based MHWs, with some events even occurring without a clear surface signature (i.e., low synchrony). The possibility that BMHW conditions could prevail with little or no surface expression has important implications for the management of marine resources, such as commercially important fisheries that live on or near the ocean bottom (e.g., lobster, crab, groundfish etc.). In particular, most widespread observing networks (including real-time monitoring systems) focus primarily on the surface ocean, with most real-time measurements coming from satellites. In very shallow regions, satellite-derived SST observations may serve as a useful proxy for bottom temperature. However, for deeper portions of continental shelves, the lack of a MHW at the surface does not necessarily indicate similarly benign bottom temperature conditions or the absence of ongoing biological impacts along the seafloor. As a result, it will be essential to maintain existing continental shelf monitoring systems and to develop new routine observational platforms (as well as ocean reanalyses) for real-time monitoring capable of alerting marine resource managers of ongoing BMHW conditions, especially when they occur without a clear surface signature.

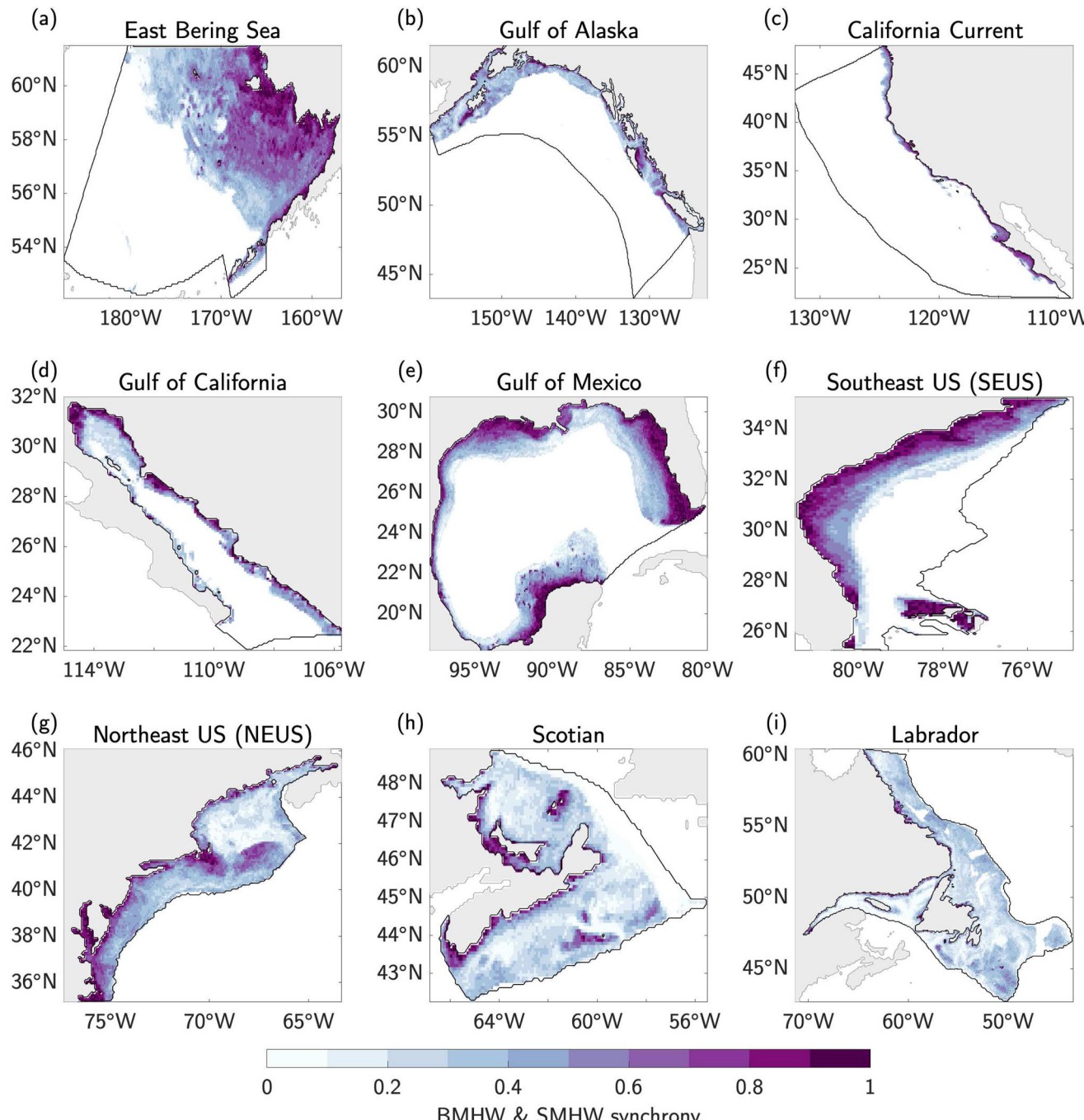

**Fig. 9 | Co-occurrence rate of bottom and surface marine heatwave events. a–i** Bottom marine heatwave (BMHW) and surface marine heatwave (SMHW) temporal synchrony as measured by the fraction of months from 1993–2019 in which BMHWs and SMHWs co-occur in each Large Marine Ecosystem (LME). A value of 1 indicates that BMHWs and SMHWs co-occur 100% of the time.

## Methods

### Bottom temperature data

To characterize BMHW events along North American coastlines, we evaluate monthly mean ocean bottom temperatures from the Global Ocean Reanalysis and Simulations 12v1 product (GLORYS)[48]. This reanalysis was generated by the Copernicus Marine Environmental Monitoring Service (CMEMS), and offers ocean variables at 1/12° horizontal resolution with 50 vertical levels. The reanalysis uses the Nucleus for European Modeling of the Ocean (NEMO) ocean model, forced at the surface by the European Center for Medium-Range Weather Forecasts (ECMWF) ERA-Interim atmospheric reanalysis. GLORYS output is available from 1993 to 2019, during which the model assimilates along-

track satellite altimetry, satellite SST, sea ice concentrations, and in situ profiles of temperature and salinity from the Coriolis Ocean database ReAnalysis (CORA) data set[48].

### Large Marine Ecosystems and depth intervals

In order to account for the varying ecological and physical regimes found throughout the North American coastal systems, we summarize BMHW characteristics for nine different LMEs surrounding the North American continent and for different depth intervals from the surface to 400 m (see Fig. 1 for each LME regions and the corresponding bathymetry). We limit our analysis of bottom temperature to ocean grid cells with bottom depths shallower than 400 m to isolate the

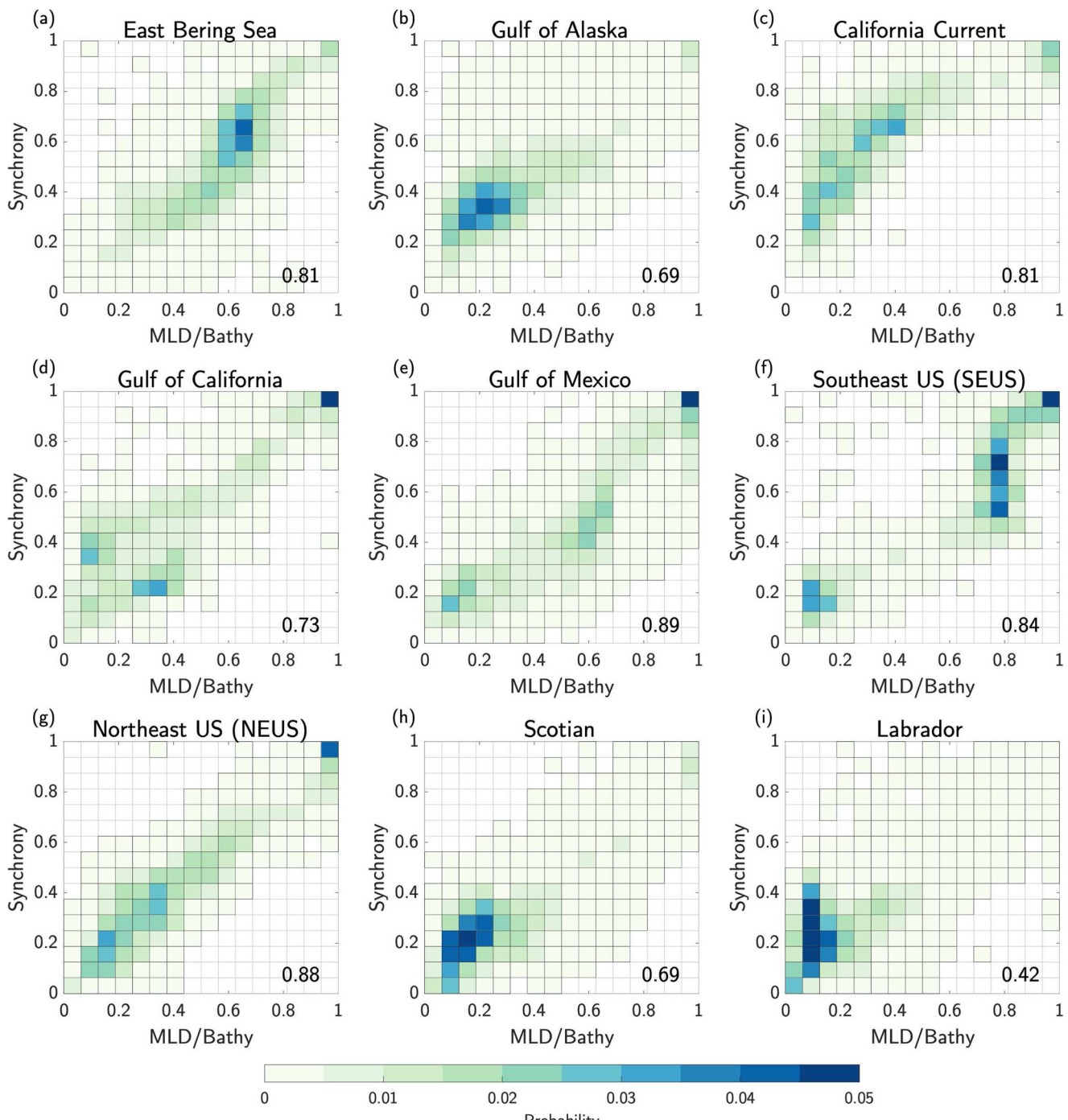

**Fig. 10 | Relationship between synchrony and mixed layer depth. a–i** Two-dimensional histograms of bottom marine heatwave (BMHW) and surface marine heatwave (SMHW) synchrony (e.g., Fig. 9) versus the ratio of mixed layer depth (MLD) to bathymetric depth (e.g., Fig. S10) composited at each grid cell when BMHW and SMHW conditions co-occurred in each Large Marine Ecosystem (LME). Shading indicates the probability that a grid cell (with bottom depth <400 m) falls within a given synchrony-MLD/bathymetry interval. For example, grid cells in the upper right quadrant represent regions within each LME with high BMHW/SMHW synchrony and a deep MLD relative to the ocean bottom. The Spearman correlation between BMHW and SMHW synchrony and MLD/bathymetry ratios across all grid cells within each LME is shown in the bottom right of each panel.

continental shelf before it slopes off to the abyssal ocean. We focus on LMEs and depth intervals (as opposed to shelf features such as width and slope) due to their relevance to marine resource management around North America.

**Justification for using GLORYS**
While the high-resolution provided by GLORYS makes it an ideal tool to investigate ocean parameters in the near-shore environment

(especially in the noticeable absence of a widespread observational network of bottom temperature), it is important to assess its fidelity in the coastal environments of North America. To this end, many studies have compared GLORYS output to independent (i.e. unassimilated) measurements of different ocean variables relevant to the spatio-temporal variability and mean state of the water column. For example, in their comparisons of different high-resolution ocean reanalyses to observations, Amaya et al.[49] and A. Castillo-Trujillo, Y.-O. Kwon, P.

Fratantoni, K. Chen, H.-Y. Seo, M. Alexander, and V. Saba (in preparation) show that GLORYS most clearly reproduces the observed mean state and variability of ocean temperature (including bottom temperature), salinity, sea surface height, and mesoscale activity along Northeast US Shelf and in the California Current System. These results are further supported by Chen et al.[35], who found that GLORYS bottom temperature is highly consistent with NEUS observations on seasonal and interannual timescales. Further, Cai et al.[50] show that GLORYS accurately reproduces the seasonal cycle and interannual variability of MLD along the Northeast US continental shelf and Amaya et al.[40] show that GLORYS realistically depicts coastally trapped waves propagating up the US west coast, from Baja to the Gulf of Alaska. Using a 0.25˚ version of GLORYS, Chi et al.[51] finds that this reanalysis stands out among other eddy-permitting products in its ability to reproduce the mean state and variability of the Gulf Stream along the Southeast US coastline. Additionally, in a comparison of global ocean reanalyses in the Benguela Current System, Russo et al.[52] found that GLORYS was the most accurate tool among those compared. Similarly, de Souza et al.[53] found that GLORYS had the smallest biases in water column structure in the nearshore environment of New Zealand. Finally, Verezemskaya et al.[54] shows that GLORYS best represents the mean and variability of temperature, salinity, and subsurface currents when compared to independent hydrographic observations of the North Atlantic. While these final three studies do not asses GLORYS in the waters surrounding North America, they provide further evidence that GLORYS is one of the most accurate ocean reanalyses available.

Many of the studies referenced above focus solely on the US west and east coasts. Therefore, we further assess GLORYS in the other regions analyzed here by comparing in situ ocean bottom temperature observations to the nearest GLORYS grid cell at ten coastal locations around North American (see Supplementary Methods and Fig. S11 and Table S1). We find that GLORYS does very well at reproducing both the general bottom temperature variability (Fig. S12) and the observed average BMHW intensity and duration (Table S2). Overall, we are highly encouraged by these comparisons, especially since these observations (to our knowledge) are not assimilated in GLORYS[48]. However, we acknowledge that, as an ocean reanalysis, GLORYS may not perfectly represent the observed evolution of subsurface temperatures, especially in regions with limited historical observations. Thus, our results may be somewhat sensitive to potential model biases and errors. Additionally, our analysis spans a relatively short time period (27 years). Therefore, it is possible that some of the statistical relationships described here may change as more data accumulates.

### Defining MHW conditions and statistical approach

We define MHW conditions (both at the surface and at the ocean bottom) using a method adapted from Hobday et al.[55] and employed by Jacox et al.[3] Using monthly mean GLORYS data, we calculate a time series of temperature anomalies relative to a 1993–2019 climatological period at every grid cell. These anomalies are then linearly detrended in order to isolate transient BMHW and SMHW events from long-term warming signals. Finally, BMHW and SMHW months are classified as periods where BWTAs and SSTAs, respectively, exceed a seasonally evolving 90th percentile threshold. For each calendar month, the 90th percentile threshold is calculated for a distribution which includes data from three consecutive months centered on the chosen month. For example, at any given grid cell, the 90th percentile threshold for January is based on a distribution made from all the December, January, and February temperature data in the record. We use this approach to generate a more robust sample for calculating the seasonally evolving 90th percentile from the relatively short GLORYS record (1993–2019), but our results and conclusions are not qualitatively affected by this choice. The spatial correlations reported here are calculated in the same manner as in Jacox et al.[3], which is based on the Spearman rank correlation coefficient and is more appropriate when comparing distributions with large positive skewness (as with MHW statistics).

Our decision to use monthly mean data is driven by the relatively slow decay rate of surface and bottom temperature along the continental shelves of North America. In particular, for most LMEs, the decorrelation timescale[43] of bottom and surface temperature anomalies is >30 days, warranting the use of monthly means (Figs. S5–S7). Nevertheless, as with many other studies on large-scale MHW events (e.g., Jacox et al.[3]), we do not expect our conclusions to be sensitive to this choice. Additionally, whether or not to linearly detrend the temperature data prior to calculating MHW statistics has been an ongoing topic of debate in the literature, with some studies noting that linearly detrending may introduce biases and errors in MHW intensity and duration for regions with strong non-linear warming trends[56]. However, upon repeating our analysis using the raw (e.g., not detrended) data, we find that the presence of warming trends has very little effect on the spatial extent or average intensity of major BMHW/SMHW events in most of the LMEs (e.g., Figs. S13, S14). This is further supported by comparing the average BMHW intensity and duration for different depth intervals in each LME for raw versus detrended data (Tables S3-S4). The clear exceptions are the Northeast US LME, the Scotian Shelf LME, and to a lesser extent the Labrador LME. These regions show greater spatial extent and SMHW/BMHW intensity from 2012-2019 for the raw data than for the detrended data (Figs. S13, S14), as well as larger average intensities and durations for nearly every depth interval (Tables S3-S4). This is not surprising given that the Northwest Atlantic Ocean has some of the strongest warming trends on the planet[57]. Regardless of these slight differences, we do not believe our decision to detrend the data ultimately changes our primary conclusions that: (1) BMHW events can have intensities and durations that exceed their surface counterparts; and (2) While SMHW and BMHW events can be connected by MLD variations, BMHW can also occur without a clear surface signature.

## Data availability

All data used in this study are available online. GLORYS reanalysis data is freely available at: https://resources.marine.copernicus.eu/products. The observational datasets used in our comparisons to GLORYS can be found at: https://www.pmel.noaa.gov/foci/data/data.html (Bering Sea - M8 and M5), http://research.cfos.uaf.edu/gak1/data/ (Gulf of Alaska - GAK1), https://www.integratedecosystemassessment.noaa.gov/regions/california-current/newport-hydrographic-line (California Current System - Newport Line), https://www.ncei.noaa.gov/access/metadata/landing-page/bin/iso?id=gov.noaa.nodc:0203749 (Gulf of Mexico - West End CP), https://www.aoml.noaa.gov/phod/wbts/data.php (Southeast US - Walton Smith), https://darchive.mblwhoilibrary.org/handle/1912/27650 (Northeast US - Martha's Vineyard Coastal Observatory), https://open.canada.ca/data/en/dataset/12184962-7879-4214-aef0-b31162f04a27 (Northeast US - Passamaquoddy Bay), https://cioosatlantic.ca/erddap/tabledap/maritimes_region_atlantic_zone_monitoring_program_rosette_vertical_profiles.subset (Scotian - Halifax Line Station 2), https://waves-vagues.dfo-mpo.gc.ca/library-bibliotheque/41063491.pdf (Labrador - Station 27). These hyperlinks can also be found in Table S1.

## Code availability

All analyses were performed using MATLAB. Codes can be accessed at https://github.com/dillon-amaya/bottom_marine_heatwave[58].

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

## Acknowledgements
We thank Tongtong Xu and Vincent Saba for their helpful comments. We further thank Melanie Fewings, Seth Danielson, Phyllis Stabeno, Emily Lemagie, Brian Dzwonkowski, Denis Volkov, Steven Lentz, Frederic Cyr, Chantelle Layton, and Hubert duPontavice for helping us procure the in situ measurements used to validate GLORYS. This research was supported in part by NOAA cooperative agreement NA22OAR4320151. C.D. and A.S.P. were supported by the NOAA Climate Program Office Modeling, Analysis, Prediction and Projection (MAPP) program NA20OAR4310378. A.C. was also supported by the NOAA MAPP program.

## Author contributions
D.J.A., M.G.J., M.A.A, and J.D.S. designed the study with input from C.D., A.C., and A.S.P. D.J.A. performed the analyses and wrote the paper with contributions from M.G.J., M.A.A, J.D.S., C.D., A.C., and A.S.P.

## Competing interests
The authors declare no competing interests.
