## [Peer Review File · Nature Communications]

Bottom marine heatwaves along the continental shelves of North AmericaREVIEWER COMMENTS

Reviewer #1 (Remarks to the Author):

The authors present the first (that I am aware of) large scale assessment of bottom marine heatwaves (BMHWs) over continental shelves using a high resolution reanalysis product that spans 27 years. The study characterizes patterns across 9 Large Marine Ecosystems (LMEs) in North America and relates them to patterns in surface MHW. The physical explanation for the patterns and characteristics leans heavily on differences in bathymetric and oceanographic complexity within and between LMEs.

The characterization of BMHWs over continental shelves and their connections to surface MHW (SMHWs) is a novel aspect of this work and the manuscript provides excellent avenues for future research into the different LME. However, I hesitate to recommend this for publication in this journal for two reasons:

Issue 1. The methodology uses reanalysis data from GLORYS, which has limited validation on continental shelves as far as I know. The authors list one study for the U.S. East coast and a second (in preparation) for the U.S. west coast, but that leaves 7 other LMEs that remain uncertain. While the authors acknowledge this potential issue, that does not stop it from being a potential issue. Certainly the 2 studies are encouraging that GLORYS can be used over continental shelves, but to me, the concern would be that by green lighting this work in such a prominent journal as Nature Communications, it would open the door to other studies that rely on reanalysis data, whose soundness is unclear. My feeling is that it will probably be good over most continental shelves and in the stated LMEs, but to me, it doesn't seem to be good practice for such a prominent journal to have such a large level of uncertainty in the primary source of information for the findings.

Issue 2. The results of this work seem a bit incremental for Nature Communications. I think the work is very good and I do see the novelty in that this is the first study to broadly examine BMHWs over large swaths of the coastal ocean. But, I still have several reservations about the novelty and/or importance of the findings.

First, several of the main findings (or aspects of them) were previously noted in Schaffer and Roughan (2017) which the authors do cite, but only in the capacity that it was a study that looked at subsurface marine heatwaves. While that study was on a much, much smaller scale, they still identify many similar ideas as this work. For example, a main finding that BMHW and SMHW become more synchronous as the mixed layer/bottom depth ratio increases is similar to the idea that "MHWs regularly extend to the bottom of the water column and are driven by downwelling favorable winds during periods of weak stratification". The finding that BMHW can be more intense than surface marine heat waves was also noted by Schaffer and Roughan (2017) as they state "... the intensity of MHWs is greatest at depth, " and they also note that "Some events even only occur at depth and would not be detected using surface temperature time series" (i.e. bottom marine heatwaves can be asynchronous with the surface).

This work unquestionably broadens these ideas, but I think it speaks to the incremental nature of the finding. They are good findings, but I just think that the findings are a bit incremental for Nature Communications. Also, I think the authors should more clearly highlight where their findings are consistent with Schaffer and Roughan (2017).

Second, the broad nature of the work inevitably leads to a limitation in explaining why we see the patterns and characteristics that are presented. At times the patterns seemed difficult to characterize leading to somewhat general comments about differences in bathymetric complexity. Most of the explanations or comments of notable events point to studies that previously identified events as surface MHW and lends on

the more in depth analysis of those citations (i.e. incremental findings). Many comments are more on the speculative side with 'may be related', 'it is possible', 'likely driven', 'suggests' peppered through the manuscript. My expectation for a paper in Nature Communication is to have more conclusive or definitive findings. Again, I think the work is good, just maybe not good enough for Nature Communications.

Other comments:

Is there anyway to strengthen analysis of the physical drivers associate with the characteristics and patterns observed. It seems like this maybe too big a task for anyone manuscript, but the conclusion that different LME have frequently have different characteristics and patterns without strongly connecting to the reasons for such differences seems to be a weakness of the manuscript to me.

Since the author lend heavily on bathymetry as a major factor into structuring the bottom marine heatwave response trends and patterns, could this be better explored by using bathymetric features like shelf wide or slope rather than by LME. As the authors note the LME approach ends up blending shelf types. This may confound response characteristics of specific shelf types. I wonder if looking at the shelf Burger number and/or some metric for stratification might further aid in the observed patterns...

Minor comments

Line 119-21 Here the Gulf of Mexico is described as wide /smooth continental shelf vs. Line 142-143 where the authors state LME of Gulf of Mexico and North American East Coast have complicated bathymetry and oceanographic features. Along those lines I think the California Current LME could be argued to have complex bathymetry (Line 119-120), e.g. capes, embayments, canyons, etc.) as well as complex oceanographic features... I am guessing that most of these LME could be argued to have complex oceanographic features. Is there a way to organize the MHW response by oceanic features. Probably not, but this leads to the somewhat unsatisfying assessment that each of this LMEs needs to be studied individually (Line 326-333).

Line 131 – I am assuming that the reporting of the R values (i.e. no significance level) is following the approach of Jacox et al. (2020), but this should be stated.

Line 149-152 – I suspect most of this LME have complex flow-bathymetry interactions so this is not a particularly satisfying explanation for the peak at ~100m.

Line 181-184 – This is reasonable, but another location where the explanation is very speculative.

Line 191-193 – Same here...

Line 191- Fewing and Brown (FMS, 2019) would be another potential reference for this event

Line 195 - – This is reasonable, but another location where the explanation is very speculative.

Line 203– This is reasonable, but another location where the explanation is very speculative.

Line 210– This is reasonable, but another location where the explanation is very speculative.

Line 212– This is reasonable, but another location where the explanation is very speculative.

Line 221– This is reasonable, but another location where the explanation is very speculative.

These instances highlight comments mentioned above. Individually they are okay, but

the accumulation of such statements lead to a general unsatisfying explanation of the patterns and characteristics observed.

Line 402- Is there a rationale for using the Jacox et al. (2020) over the Hobday et al. (2016)? What should be the guiding principle for this selection? I am guessing the motivation will be similar to that of Jacox et al. (2020) which should be briefly mentioned. Over the shelf where temporal and spatial scales are reduced relative to the open ocean one might argue that the 5-day methodology of Hobday et al. (2016) might be more appropriate. Maybe it doesn't make much of a difference? Regardless, this does raise the question about when/why to use different ways of defining marine heatwaves. This could be a discussion for a different paper though.

Line 410 What is the reasoning of the 90th percentile from the 3 month distribution as opposed to just the monthly distribution? Seems like this could artificially bias the distribution of marine heatwave (e.g., peak coolest/warmest months will have lower values in the distributions used to calculate the 90th percentile?)

Figure 4 – Colorbar has two shades in 0-1 month range, but with monthly data, I am not sure how a marine heatwave could last less than 1 month (and I don't see any colors in the shades). Maybe change the color bar to remove the 0-1 range.

Reviewer #2 (Remarks to the Author):

- Summary

- Using the GLORYS 1/12° model output (1993-2019) the authors have calculated marine heatwaves (MHW) at the bottom (BHW) and the surface (SHW) in the North American LME for grid pixels < 400 m deep in order to 1) document the intensity and duration of BHW and 2) compare them to SHW. As the body of literature on SHW is very well established, the authors are able to mainly focus on the BHW results. With SHW being useful here for comparison.

- What are the noteworthy results?

- The authors show some very interesting differences in the intensity and the spatial/temporal occurrence of BHW and SHW both within and across LME. Within LME by showing how there can be time lags between BHW and SHW, or even the occurrence of one without the other. Between LME by showing how the neighbouring LME can experience similar BHW and/or SHW, often with a time delay or other notable difference. This is attributed largely to the shallowness of a given pixel in that these are more likely to have a MLD that can extend to the bottom. The authors also provide an interesting discussion on areas where this relationship is less linear, likely due to areas with more bathymetric variability. An interesting area of research in the future to be sure.

- Will the work be of significance to the field and related fields? How does it compare to the established literature? If the work is not original, please provide relevant references.

- The work will certainly be of significance to the field of MHW studies and to related fields that study these events in physical oceanography and ecology. And likely many more (e.g. demersal fisheries). It compares very well to the established literature and is an obvious and much needed progression of it. There have been many other studies that look at BHW in some capacity, but none at this scale, and the authors reference these studies here.

- Does the work support the conclusions and claims, or is additional evidence needed?

- The work here definitely supports the conclusions/claims made, but I found the paper to be a bit light on them considering how impressive the results presented here are. Looking at Figure 8 in particular there are dozens of features whose investigations could require an entire fleet of researchers to follow up on, which would in turn be very interesting and important contributions to the literature. That being said, by not making

any claims about the development of BHW over time the authors have avoided any complications from the one issue I have with the analysis.

- Are there any flaws in the data analysis, interpretation and conclusions? Do these prohibit publication or require revision?

- An ongoing topic of debate in the field of MHW study is whether or not to first de-trend time series before detecting events, with the authors using here a well established de-trending methodology. The argument between the two being (which I agree with) that de-trending better allows an investigator to isolate the historic/present drivers of extreme events independent of the long-term climate change signal. A paper however has recently been published (see specific comments below for reference/link) that quantifies the error that linear de-trending introduces into MHW results. Notably that it tends to over-inflate the intensity/duration of events at the beginning of time series because the rate of warming has generally been faster over the last decade. This also tends to cause a dip in the duration/intensity of events in the middle of the time series, as one may see in the occurrence of BHW in the E Bering Sea in Figure 5. I think that the analysis as it is now is fantastic and nothing should be changed, but that the manuscript would benefit from one or two sentences, either in the methods or discussion section, that addresses this potential issue. Another thought I had while reading this was that rates of warming between the bottom and the surface are likely to be different for much of study areas here. Meaning that de-trending the time series before detecting and comparing results will likely introduce different biases (of course, so would not de-trending the data) at the bottom and surface. I would be interested to read what the authors thought of that potential issue. As they see fit to do so.

- Is the methodology sound? Does the work meet the expected standards in your field?

- Yes, the methodology is well established and the work is at the highest level of quality in the field.

- Is there enough detail provided in the methods for the work to be reproduced?

- More or less, yes. It's not an issue of detail, but rather computing power, in reproducing the results. But that certainly is not the burden of the authors. Depending on the current publication requirements of Nature, I think that the MATLAB code used for the analysis should be uploaded to a public repository, rather than "Scripts are available upon requests."

All the best,

-Robert Schlegel

- Specific comments:

- Abstract:

- As I think that most non-physical oceanographers that read this will assume that these results are based on time series with the climate change signal not removed (i.e. non-de-trended), please somewhere point out to the reader that these results are based on de-trended time series.

- Intro:

- Lns 59-67: As the authors prefer, consider adding Darmaraki et al. (2019) to this list of BHW analyses to include the Mediterranean. I am not a co-author.

- Ln 73: "BTAs" here and throughout; I've noticed a trend in the heatwave literature where authors have begun no longer including the "s" after the plural form of acronyms. I think it looks a bit cleaner to remove the "s". Completely at the authors /editors discretion.

- Ln 93: "high-resolution"; I appreciate that this section is meant to be as direct as possible, but I think many readers would like to see the resolution of the model included here in parentheses. As the editing process prefers.

- Results:

- Ln 109: "NEUS" here and throughout; should this be "NE US"? It took me a moment to realise where this was as I initially thought it was an acronym for some climate index or other physical force. Same for "SEUS" etc.

- Lns 114-123: Very interesting insight that I hadn't realised just from looking at the figures beforehand.

- Ln 126: “scatter diagrams” → “scatter plots”
- Lns 144-152: Also very interesting insight.
- Ln 153: “varies strongly within and among” This phrase implies a statistical test (e.g. two way ANOVA) was performed.
- Lns 166-168: Creating a spatial average of an LME does not show whether or not events co-occurring in time within it are spatially cohesive. Meaning that one can have many small patches of events caused by different forces. Though this is more of a surface signature, and I doubt that it is occurring at depth.
- Lns 179-181: Indeed, when one linearly detrends a time series in a region highly influenced by ENSO, it is the Nino3.4 index that best correlates to the heatwave statistics.
- Lns 201-204: While it may be outside of the scope of this article, one could certainly devise a methodology that uses the results here in which one could conclude whether the events are spatially consistent or not. Perhaps an area of future research. I think it would be interesting in the LME with greater geomorphological variance to show the spatial differences.
- Ln 211: “Chen et al. 2021” → “Chen et al., 2021”
- Lns 216-225: The north-south difference is interesting.
- Lns 226-227: That’s surprising that spatial surface intensity is less variable than bottom. Or does this mean that SHW in a given LME tend to be the same event, so have a similar signal, whereas BHW can be different events, therefore having more variance?
- Lns 227 and 229: Switch the references to Figure S3 and S4.
- Lns 227-229: This is difficult to understand.
- Lns 285-287: Good to know.
- Lns 294-297: I think it would be good to provide a condensed form of this explanation to the caption for Figure 10. Having looked at the figures before reading the manuscript, it took me a moment to be certain which direction the X-axis was going.
 - Summary and Discussion
- Ln 301: “high-resolution ($1/12^\circ$)” Very nice. I’ve considered using this product before. But the size of the data... jikes.
- Lns 327-330: I’m not convinced, given the crude nature of the spatial aggregation, that the authors can make this concluding statement.
- Lns 330-333: Certainly an interesting future avenue of study.
- Lns 334-350: The authors may want to remove some of this corresponding text from the results section as much of this has already been stated there.
- Lns 368-371: This would be amazing to have.
 - Methods:
 - Lns 404-405: While for the last few years the disagreement about de-trending vs not de-trending time series before detecting MHW has been a philosophical one (based largely on whether or not the field of research in question is physical oceanography or ecology), there is now a publication that quantifies the error introduced into MHW results from linear de-trending time series. Please see Wang et al. (2022: https://journals.ametsoc.org/view/journals/atot/aop/JTECH-D-21-0142.1/JTECH-D-21-0142.1.xml?tab_body=pdf) for the specific statistical models used etc. That being said, the methodology used here is indeed well established, so I am not suggesting that anything be recalculated. Rather, I think that the authors should include one or two sentences explaining to the reader how the method of linear de-trending inflates the intensity of MHW earlier in a time series when linearly de-trended.
 - Code availability
 - “Scripts are available upon requests.” Does not mean that the code is open access. It is best practice to list the GitHub (or otherwise) repo where the code is hosted.
 - Figure 1:
 - A good start to the figures and useful to have.
 - Figure 2:
 - Also useful, but a bit unfortunate about the California Current panel having so much white space. No recommended changes though.
 - Figure 3:
 - I like these 2-D histograms a lot. The grey dots are a useful addition.
 - Figure 4:

- If the methodology analyses monthly data, how can the average duration of an event last for 0 or 0.5 months? Shouldn't then this (and other duration colourbars throughout) start at 1, rather than 0?
- Figure 5:
- These results are mesmerising.
- Figure 6:
- The thin shelf edge / Gulf Stream signal in panel g for intensity differences is very interesting.
- Figure 7:
- And how it's not present here for the duration.
- Figure 8:
- Overlaying the BHW and SHW results here is incredible. It is of course outside the scope of this paper, but an interactive website to explore these results would be awesome.
- Figure 9:
- I find this to be the most interesting of all of the map based figures. It does a great job of showing how different the bottom-surface relationship is over The Grand Banks than almost all LME.
- Figure 10:
- I'm not completely sure what the shading in this figure is showing. Perhaps explain it again in the caption in more detail.
- Figure S1:
- Future work could cut the LME into smaller areas to see which of the upward curves of dots that emerge from the cloud (as in panel c) relate to which smaller regions.
- Figure S2:
- Interesting.
- Figure S3:
- A lot of interesting relationships to unpack here in the future.
- Figure S4:
- Helpful for understanding the referencing statement in the manuscript.
- Figure S5:
- This is another very interesting way of comparing surface and bottom temperature anomalies.
- Figure S6:
- I find the two different statistical comparisons here very useful. Why not add these to the figures in the main manuscript?
- Figure S7:
- This does a good job of explaining the patterns in Figure 9. Perhaps reference this figure in that caption.

Reviewer #1 (Remarks to the Author):

The authors present the first (that I am aware of) large scale assessment of bottom marine heatwaves (BMHWs) over continental shelves using a high-resolution reanalysis product that spans 27 years. The study characterizes patterns across 9 Large Marine Ecosystems (LMEs) in North American and relates to them to patterns in surface MHW. The physical explanation for the patterns and characteristics leans heavily on differences in bathymetric and oceanographic complexity within and between LMEs.

- Thank you for the time and effort you put into your review of our manuscript. You will find responses to your individual comments below. Note that some of the comments have been reordered so as to better group by similar topic. Doing so allows us to better respond to each of the broader concerns raised. Please also note that Line and Figure references in our responses are for the *revised* manuscript.

The characterization of BMHWs over continental shelves and their connections to surface MHW (SMHWs) is a novel aspect of this work and the manuscript provides excellent avenues for future research into the different LME. However, I hesitate to recommend this for publication in this journal for two reasons:

Issue 1. The methodology uses reanalysis data from GLORYS, which has limited validation on continental shelves as far as I know. The authors list one study for the U.S. East coast and a second (in preparation) for the U.S. west coast, but that leaves 7 other LMEs that remain uncertain. While the authors acknowledge this potential issue, that does not stop it from being a potential issue. Certainly the 2 studies are encouraging that GLORIES can be used over continental shelves, but to me, the concern would be that by green lighting this work in such a prominent journal as Nature Communication, it would open the door to other studies that rely on reanalysis data, whose soundness is unclear. My feeling is that it will probably be good over most continental shelves and in the stated LMEs, but to me, it doesn't seem to be good practice for such a prominent journal to have such a large level of uncertainty in the primary source of information for the findings.

- We appreciate your concerns regarding the fidelity of GLORYS. There are several reasons we feel that GLORYS is appropriate for high-impact studies (i.e., studies published by the *Nature* family of journals), including new analyses we have conducted to help address these concerns:
 1. In order to further validate GLORYS bottom temperature variability around North America we have conducted a new comparison of GLORYS to several *in situ* measurements of long-term bottom temperature within the different LMEs discussed in this paper (see Figures R1-R2 and Table R1 below). In short, our new comparisons indicate that GLORYS accurately reproduces observed bottom temperature variability and BMHW statistics around North America. A detailed description of these results can be found in the following section of this response letter entitled "Observational comparisons to GLORYS bottom temperature".

2. In addition to our brand-new observational comparisons with GLORYS bottom temperature, GLORYS routinely outperforms its peers (i.e., other publicly available global ocean reanalyses) when compared to raw observations of other ocean variables. In their comparisons of different high-resolution ocean reanalyses to observations, Chen et al. (2021), Amaya et al. (*in review*), and Castillo-Trujillo et al. (*in review*) show that GLORYS most clearly reproduces the observed mean state and variability of ocean temperature (including bottom temperature), salinity, sea surface height, and mesoscale activity along Northeast US Shelf and in the California Current System. Further, Cai et al. (2021) show that GLORYS accurately reproduces the seasonal cycle and interannual variability of MLD along the Northeast US continental shelf and Amaya et al. (2022) show that GLORYS realistically depicts coastally trapped waves propagating up the US west coast, from Baja to the Gulf of Alaska. Using a 0.25° version of GLORYS, Chi et al. (2018) finds that this reanalysis stands out among other eddy-permitting products in its ability to reproduce the mean state and variability of the Gulf Stream along the Southeast US coastline. Additionally, in a comparison of global ocean reanalyses in the Benguela Current System, Russo et al. (2022) found that GLORYS was the most accurate tool among those compared. Similarly, de Souza et al. (2021) found that GLORYS had the smallest biases in water column structure in the nearshore environment of New Zealand. Finally, Verezhenskaya et al. (2021) shows that GLORYS best represents the mean and variability of temperature, salinity, and subsurface currents when compared to independent hydrographic observations of the North Atlantic. While these final three studies do not assess GLORYS in the waters surrounding North America, they provide further evidence that GLORYS is one of the most (if not the most) accurate ocean reanalysis available.
3. There are few alternative datasets to conduct an analysis such as ours. In order to resolve the narrow shelf regions surrounding North America, we require horizontal grid resolutions of less than 25km, ruling out many potential gridded datasets that include physically consistent measurements of surface temperature, bottom temperature, and MLD. Regional ocean models with limited domains that cover each LME could be an option, however, these regional model experiments do not already exist for many of the LMEs (or at least are not publicly available). Given the breadth of our analysis, running these simulations ourselves is not computationally realistic. Plus, it is not clear what benefit regional ocean models would add over a global data assimilative product such as GLORYS, especially since: (1) Regional ocean models suffer from uncertainties in their lateral boundaries and surface forcing; and (2) GLORYS horizontal resolution is the roughly the same as typical regional simulations, which often have resolutions of 7-10 km (e.g., Neveu et al. 2016). Pure observations are not an option since they are comparably noisy, discontinuous, sporadic, and often poorly documented, especially for measurements of ocean bottom temperature. Additionally, there simply are no widespread bottom temperature measurements with which to derive BMHW statistics in time and space. There are some long-term point measurements (see our new comparisons below), but the only systematic bottom temperature observations that we are aware of come from fish trawls. However, these measurements are randomly stratified in space and not consistent in time, thus limiting their use for assessing bottom temperature variability.

4. Finally, ocean reanalyses have been the focal point of many important studies published in the *Nature* family of journals (see among others, Morim et al. 2022; Huguenin et al. 2022; Li et al. 2022; Shu et al. 2021; Wang et al. 2021; Wengel et al. 2021; Hayashi et al. 2020; Amaya et al. 2020), many of which have even used the same GLORYS dataset analyzed here (Chambault et al. 2020; Etourneau et al. 2019; Sgubin et al. 2017). Therefore, the use of reanalyses in high-impact publications is not without precedent.

In summary, the GLORYS ocean reanalysis is state-of-the-art and provides a novel dataset which allow us, for the first time, to assess bottom temperature variability across multiple scales. An analysis of this scope would not be possible without such a tool. In response to Issue 1, we have added our new observational comparisons to the Supplementary Information (Figures S8-S9, Tables S1-S2) and we have summarized the results (which are discussed in the following section) in the Methods. We have also added text to the Methods to further justify our use of GLORYS based on the points above.

Observational comparisons to GLORYS bottom temperature

As mentioned above, *in situ* measurements of bottom temperature along North America's continental shelves are sporadic, with very few examples of observations being collected at the same location for long enough to assess bottom temperature variability on climate-relevant timescales. Nevertheless, there are some limited long-term measurements of bottom temperature in the different LMEs with which to compare to GLORYS. Please note that we went to great lengths to gather these observations, which came from many different sources with many different formats and guidelines. Aggregating these observations was challenging due to the lack of a common repository for the storage and documentation of long-term observations across LMEs. As a result, the difficulty that we experienced procuring these observations is further testament to the benefit of a tool like GLORYS.

Figure R1 shows the locations of the ten long-term bottom temperature observational networks used in our comparisons. Each LME is represented, with the exception of the Gulf of California where we could not find long-term bottom temperature observations. Observational platforms used to make these measurements range from long-term moorings (BS-M8, BS-M5, GAK1, and West End CP), repeat hydrographic sections (Newport Line, Walton Smith, Passamaquoddy Bay, Halifax Line and Station 27), and an undersea cable (MVCO). For a description of each of these observations, see the new "Observational comparisons to GLORYS" section in the Supplementary Information.

Figure R2 compares monthly anomalies of near-bottom temperature between the observations and the nearest GLORYS grid cell. However, the repeat hydrographic sections from the Walton Smith and Halifax Line locations are conducted ~quarterly, and thus it was not possible to calculate monthly anomalies for these locations. Instead, we show the full near-bottom temperature values from each CTD cast and compare those to GLORYS (Figure R2f,i). At each location, GLORYS compares favorably to the *in situ* point measurement, with R-values ranging from 0.53 to as high as 0.84. Note that while we compare the observations at each location with the nearest GLORYS grid cell (see Supplementary Information), there are instances where the observed data and GLORYS

data are measured several kilometers apart. Thus, in some locations, like those with rapidly changing bathymetry, differences between GLORYS and observations may result from these slight differences in location rather than from errors in the model. Therefore, we do not expect perfect agreement between observations and the reanalysis. Nevertheless, all R-values are significant at the 99% confidence level.

We further compare GLORYS to the observations by calculating BMHW average intensity and duration at each location where it was possible to derive monthly anomalies (Table R1). The average BMHW statistics calculated from GLORYS and the co-located observations are quite consistent, with intensity differences rarely exceeding a few tenths of a degree Celsius and duration differences rarely exceeding half a month. However, there are some larger differences at BS-M5 and to a lesser extent at GAK1.

While we acknowledge that these observational comparisons are based on point measurements and may not represent the full range of bottom temperature variability found within a given LME, we are highly encouraged by how well GLORYS captures these observations, especially since these measurements (to our knowledge) are not assimilated in GLORYS (Lellouche et al. 2021). Therefore, based on these results, as well as the growing body of research that have already indicated GLORYS accuracy (see above references), we are confident that GLORYS is a trustworthy tool for our analysis, especially given the noticeable absence of a widespread observational network of bottom temperature.

References

- Amaya, D. J., Miller, A. J., Xie, S.-P., & Kosaka, Y. (2020). Physical drivers of the summer 2019 North Pacific marine heatwave. *Nature Communications*, *11*(1), 1903. <https://doi.org/10.1038/s41467-020-15820-w>
- Amaya, D. J., Jacox, M. G., Dias, J., Alexander, M. A., Karnauskas, K. B., Scott, J. D., & Gehne, M. (2022). Subseasonal-to-Seasonal Forecast Skill in the California Current System and Its Connection to Coastal Kelvin Waves. *Journal of Geophysical Research: Oceans*, *127*(1), e2021JC017892. <https://doi.org/10.1029/2021JC017892>
- Cai, C., Kwon, Y.-O., Chen, Z., & Fratantoni, P. (2021). Mixed layer depth climatology over the northeast U.S. continental shelf (1993–2018). *Continental Shelf Research*, *231*, 104611. <https://doi.org/10.1016/j.csr.2021.104611>
- Chambault, P., Tervo, O. M., Garde, E., Hansen, R. G., Blackwell, S. B., Williams, T. M., et al. (2020). The impact of rising sea temperatures on an Arctic top predator, the narwhal. *Scientific Reports*, *10*(1), 18678. <https://doi.org/10.1038/s41598-020-75658-6>
- Chen, Z., Kwon, Y. O., Chen, K., Fratantoni, P., Gawarkiewicz, G., Joyce, T. M., et al. (2021). Seasonal Prediction of Bottom Temperature on the Northeast U.S. Continental Shelf. *Journal of Geophysical Research: Oceans*, *126*(5). <https://doi.org/10.1029/2021JC017187>
- Chi, L., Wolfe, C. L. P., & Hameed, S. (2018). Intercomparison of the Gulf Stream in ocean reanalyses: 1993–2010. *Ocean Modelling*, *125*, 1–21. <https://doi.org/10.1016/j.ocemod.2018.02.008>

- Etourneau, J., Sgubin, G., Crosta, X., Swingedouw, D., Willmott, V., Barbara, L., et al. (2019). Ocean temperature impact on ice shelf extent in the eastern Antarctic Peninsula. *Nature Communications*, *10*(1), 304. <https://doi.org/10.1038/s41467-018-08195-6>
- Hayashi, M., Jin, F.-F., & Stuecker, M. F. (2020). Dynamics for El Niño-La Niña asymmetry constrain equatorial-Pacific warming pattern. *Nature Communications*, *11*(1), 4230. <https://doi.org/10.1038/s41467-020-17983-y>
- Huguenin, M. F., Holmes, R. M., & England, M. H. (2022). Drivers and distribution of global ocean heat uptake over the last half century. *Nature Communications*, *13*(1), 4921. <https://doi.org/10.1038/s41467-022-32540-5>
- Lellouche, J.-M., Greiner, E., Bourdallé Badie, R., Garric, G., Melet, A., Drévilion, M., et al. (2021). The Copernicus Global 1/12° Oceanic and Sea Ice GLORYS12 Reanalysis. *Frontiers in Earth Science*, *9*. <https://doi.org/10.3389/feart.2021.698876>
- Li, Z., Ding, Q., Steele, M., & Schweiger, A. (2022). Recent upper Arctic Ocean warming expedited by summertime atmospheric processes. *Nature Communications*, *13*(1), 362. <https://doi.org/10.1038/s41467-022-28047-8>
- Morim, J., Erikson, L. H., Hemer, M., Young, I., Wang, X., Mori, N., et al. (2022). A global ensemble of ocean wave climate statistics from contemporary wave reanalysis and hindcasts. *Scientific Data*, *9*(1), 358. <https://doi.org/10.1038/s41597-022-01459-3>
- Neveu, E., Moore, A. M., Edwards, C. A., Fiechter, J., Drake, P., Crawford, W. J., et al. (2016). An historical analysis of the California Current circulation using ROMS 4D-Var: System configuration and diagnostics. *Ocean Modelling*, *99*, 133–151. <https://doi.org/10.1016/j.ocemod.2015.11.012>
- Russo, C. S., Veitch, J., Carr, M., Fearon, G., & Whittle, C. (2022). An Intercomparison of Global Reanalysis Products for Southern Africa’s Major Oceanographic Features. *Frontiers in Marine Science*, *9*. Retrieved from <https://www.frontiersin.org/articles/10.3389/fmars.2022.837906>
- Sgubin, G., Swingedouw, D., Drijfhout, S., Mary, Y., & Bennabi, A. (2017). Abrupt cooling over the North Atlantic in modern climate models. *Nature Communications*, *8*(1), 14375. <https://doi.org/10.1038/ncomms14375>
- Shu, Q., Wang, Q., Song, Z., & Qiao, F. (2021). The poleward enhanced Arctic Ocean cooling machine in a warming climate. *Nature Communications*, *12*(1), 2966. <https://doi.org/10.1038/s41467-021-23321-7>
- de Souza, J. M. A. C., Couto, P., Soutelino, R., & Roughan, M. (2021). Evaluation of four global ocean reanalysis products for New Zealand waters—A guide for regional ocean modelling. *New Zealand Journal of Marine and Freshwater Research*, *55*(1), 132–155. <https://doi.org/10.1080/00288330.2020.1713179>
- Verezemskaya, P., Barnier, B., Gulev, S. K., Gladyshev, S., Molines, J.-M., Gladyshev, V., et al. (2021). Assessing Eddyding (1/12°) Ocean Reanalysis GLORYS12 Using the 14-yr Instrumental Record From 59.5°N Section in the Atlantic. *Journal of Geophysical Research: Oceans*, *126*(6), e2020JC016317. <https://doi.org/10.1029/2020JC016317>
- Wang, J., Church, J. A., Zhang, X., & Chen, X. (2021). Reconciling global mean and regional sea level change in projections and observations. *Nature Communications*, *12*(1), 990. <https://doi.org/10.1038/s41467-021-21265-6>
- Wengel, C., Lee, S.-S., Stuecker, M. F., Timmermann, A., Chu, J.-E., & Schloesser, F. (2021). Future high-resolution El Niño/Southern Oscillation dynamics. *Nature Climate Change*, *11*(9), 758–765. <https://doi.org/10.1038/s41558-021-01132-4>

Figure R1 Location of different long-term near-bottom temperature observations used to compare with GLORYS. Shading denotes ocean bottom depth (meters).

Figure R2 Timeseries of near-bottom temperature (NBT; °C) at the different locations shown in Figure R1. Observations are in black and GLORYS is in red. All values are monthly mean NBT anomalies except for the Walton Smith and Halifax Line locations, which are based on ~quarterly NBT measurements. The correlation coefficient (R-value) between observations and GLORYS at each location is shown in the title. All R-values are significant at the 99% confidence level.

Table R1 Average BMHW intensity (°C) and duration (months) at each location shown in Figure R1. Values are calculated based on linearly detrended and raw (e.g., not detrended) monthly anomalies. Black numbers indicate values calculated using observations and red numbers indicate values calculated using the nearest GLORYS grid cell.

Locations	Average Intensity (°C)				Average Duration (months)			
	Raw		Detrended		Raw		Detrended	
Bering Sea (BS-M8)	1.9	1.8	1.6	1.4	2.7	3.2	2.7	2.7
Bering Sea (BS-M5)	3.0	1.8	2.3	1.6	5.3	4.0	4.0	3.2
G. of Alaska (GAK1)	0.8	0.6	0.7	0.6	3.3	2.3	2.9	3.3
CCS (Newport Line)	1.9	2.2	1.8	2.2	1.3	1.6	1.5	2.0
GOM (West End CP)	2.1	2.0	2.1	2.0	1.3	1.3	1.3	1.4
NEUS (MVCO)	2.1	2.0	2.1	1.9	1.9	1.5	1.7	1.7
NEUS (Passam. Bay)	1.7	1.4	1.5	1.2	1.6	1.6	1.4	1.5
Lab. Shelf (Station 27)	0.8	0.9	0.8	0.9	3.0	3.0	3.0	2.7

Issue 2. The results of this work seem a bit incremental for Nature Communications. I think the work is very good and I do see the novelty in that this is the first study to broadly examine BMHWs over large swaths of the coastal ocean. But, I still have several reservations about the novelty and/or importance of the findings.

First, several of the main finding (or aspects of them) were previous noted in Schaffer and Roughan (2017) which the authors do cite, but only in the capacity that it was a study that looked at subsurface marine heatwaves. While that study was on a much, much smaller scale, they still identify many similar ideas as this work. For example, a main finding that BMHW and SMHW become more synchronous as the mixed layer/bottom depth ratio increase is similar to the idea that “MHWs regularly extend to the bottom of the water column and are driven by downwelling favorable winds during periods of weak stratification”. The finding that BMHW can be more intense than surface marine heat waves was also noted by Schaffer and Roughan (2017) as they state “ ... the intensity of MHWs is greatest at depth, “ and they also not that “Some events even only occur at depth and would not be detected using surface temperature time series” (i.e. bottom marine heatwaves can be asynchronous with the surface).

This work is unquestionably broadens these ideas, but I think it speaks to the incremental nature of the finding. They are good finding, but I just think that the findings are a bit incremental for Nature Communications. Also, I think the authors should more clearly highlight where there finding are consistent with Schaffer and Roughan (2017).

- We agree that similar themes emerge from our study and from Schaeffer and Roughan (2017), however, we disagree with the perceived incremental nature of our findings. In particular, we point to three areas which our study significantly expands on Schaeffer and Roughan (2017). (1) Although we investigate a similar hypothesis regarding the role of stratification/MLD in connecting surface and subsurface MHWs, we are able to examine the extent which these physical processes hold true in general (i.e., over much larger spatial scales and across more varied physical environments); (2) We examine the spatial footprint of bottom and surface MHW events and how they may be tied to different bathymetric features and depth ranges for different shelf systems, allowing us to identify key features for future investigation; and (3) We compare events across different regions in space and time to identify potential connections in their spatial evolution and timing (e.g., the lagged relationship in the 1997/1998 BMHW events from the Gulf of California to the Gulf of Alaska).

In response to this concern, we have added several sentences to the final paragraph of the Summary and Discussion section to more clearly outline how our results build upon earlier work by Schaeffer and Roughan (2017).

Second, the broad nature of the work inevitably leads to a limitation in explaining why we see the patterns and characteristics that are presented. At times the patterns seemed difficult to characterized leading to somewhat general comments about differences in bathymetric complexity. Most of the explanations or comments of notable events point to studies that previously identified events as surface MHW and lends on the more in-depth analysis of those citations (i.e. incremental findings). Many comments are more on the speculative side with ‘may be related’, ‘it is possible’,

'likely driven', 'suggests' peppered through the manuscript. My expectation for a paper in Nature Communication is to have more conclusive or definitive findings. Again, I think the work is good, just maybe not good enough for Nature Communications.

Other comments:

Is there any way to strengthen analysis of the physical drivers associated with the characteristics and patterns observed. It seems like this maybe too big a task for anyone manuscript, but the conclusion that different LME have frequently have different characteristics and patterns without strongly connecting to the reasons for such differences seems to be a weakness of the manuscript to me.

Line 149-152 – I suspect most of this LME have complex flow-bathymetry interactions so this is not a particularly satisfying explanation for the peak at ~100m.

Line 181-184 – This is reasonable, but another location where the explanation is very speculative.

Line 191-193 – Same here

Line 195 – This is reasonable, but another location where the explanation is very speculative.

Line 203 – This is reasonable, but another location where the explanation is very speculative.

Line 210 – This is reasonable, but another location where the explanation is very speculative.

Line 212 – This is reasonable, but another location where the explanation is very speculative.

Line 221 – This is reasonable, but another location where the explanation is very speculative.

These instances highlight comments mentioned above. Individually they are okay, but the accumulation of such statements leads to a general unsatisfying explanation of the patterns and characteristics observed.

- We have aggregated the Reviewer comments above since they are all related to the same topic and we will respond to their broader point here. We sincerely appreciate the desire to feature a more detailed analysis of the physical drivers of BMHW events in each LME, and we have done our best to balance this preference with the realistic limitations of our single study. For example, our investigation of the relationship between SMHW, BMHW, and the MLD was a sincere effort to provide a level of physical understanding that was broadly applicable to the dynamically diverse regions we study in our manuscript. While we would like to have also investigated the granular mechanisms that govern BMHW evolution in each region, as you have alluded, such an effort would be beyond the scope of a single paper. Indeed, we are limited by *Nature Communications* to 10 display items (figures + tables), and we believe that the 10 figures we already have in the main text are essential to drawing the primary conclusions of our study. Even if we were allotted more space (say in a different journal), we would still be hesitant to expand our analysis into the physical mechanisms of each region. A paper of that scope would need to either be unreasonably long (e.g., with dozens of figures), or exceedingly brief in its description of the relevant physical drivers in each region. Neither of these approaches would benefit the marine resource management or scientific communities, thus we believe that each of these LMEs requires a dedicated focus.

In lieu of a detailed analysis of the unique mechanisms that govern BMHWs in each LME, we outline a number of reasonable physical hypotheses in the Summary and Discussion section for what could be driving the different BMHW characteristics. These hypotheses, as you point out, are necessarily speculative; however, they are informed by our (the co-authors) collective understanding of ocean dynamics and supported by peer-reviewed literature, thus giving them a firm physical basis. We believe that our results and the physical mechanisms they suggest provide critical insight that will inspire and guide future exploration into these topics. To that end, we have already begun a separate detailed analysis of the different mechanisms leading to BMHW events in the California Current System (CCS). Preliminary results indicate that the physical drivers of BMHWs in the CCS are multi-faceted and complex, with thermocline displacements driving BMHWs from Baja California to the Southern California Bight and upwelling variability producing BMHWs further north. Additionally, we are collaborating with researchers at GFDL to quantify the mechanisms behind BMHWs along the Northeast US continental shelf.

In an effort to provide additional physical context while also adhering to the length limitations of *Nature Communications*, we have conducted a new analysis of the decorrelation timescales of surface and bottom temperature anomalies in each LME (see Figures R3-R5 and Table R2 below in our response to your comment about daily versus monthly means). These new results allow us to better understand the differences between the SMHW and BMHW duration patterns, and further supports our previous assumptions about the relationship between SMHW events and air-sea fluxes (see Lines 208-213).

As to the perceived incremental nature of our work, our comprehensive analysis of BMHW events around North America is the first of its kind, significantly building on previous examples that only focus on single point measurements (e.g., Schaeffer and Roughan, 2017). This breakthrough in scale and understanding has been enabled by a state-of-the-art observational tool—the GLORYS ocean reanalysis—which is being analyzed here from a novel perspective. Our results provide decision makers with critical insight into the characteristics of BMHW events, including where and when these conditions have been associated with SMHW events. The possibility that BMHW conditions could prevail with little or no surface expression has clear implications for the management of marine resources, such as commercially important fisheries that live on or near the ocean bottom (e.g., lobster, crab, groundfish etc.). In particular, most widespread observing networks (including real-time monitoring systems) focus primarily on the surface ocean, with most real-time measurements coming from satellites. In very shallow regions, satellite-derived SST observations may serve as a useful proxy for bottom temperature. However, our results indicate that for deeper portions of continental shelves, the lack of a MHW at the surface does not necessarily indicate similarly benign bottom temperature conditions or the absence of ongoing biological impacts along the seafloor.

For these reasons, we believe our results and conclusions meet the standards of this journal and are of wide interest to both the marine resource management and scientific communities. In response to these comments, we have added text to the Discussion section to further highlight the novel aspects of our study.

Since the author lend heavily on bathymetry as a major factor into structuring the bottom marine heatwave response trends and patterns, could this be better explored by using bathymetric features like shelf wide or slope rather than by LME. As the authors note the LME approach ends up blending shelf types. This may confound response characteristics of specific shelf types. I wonder if looking at the shelf Burger number and/or some metric for stratification might further aid in the observed patterns...

Minor comments

Line 119-21 Here the Gulf of Mexico is described as wide/smooth continental shelf vs. Line 142-143 where the authors state LME of Gulf of Mexico and North American East Coast have complicated bathymetry and oceanographic features. Along those lines I think the California Current LME could be argued to have complex bathymetry (Line 119-120), e.g. capes, embayments, canyons, etc.) as well as complex oceanographic features...I am guessing that most of these LME could be argued to have complex oceanographic features. Is there a way to organize the MHW response by oceanic features. Probably not, but this leads to the somewhat unsatisfying assessment that each of this LMEs needs to be studied individually (Line 326-333).

- We have again combined these comments since they concern a similar topic. Our choice to focus on Large Marine Ecosystems (as opposed to shelf types) is partly because they provide convenient boundaries between different large-scale dynamical regimes. For example, the California Current LME is dominated by an equatorward eastern boundary current, and its borders help to separate it from regions like the Gulf of Alaska LME, which is dominated by the poleward Alaska Current. If we were instead to focus on shelf type, rather than geographical location, we would end up blending events across very different dynamical regimes (by say, lumping all events that occurred on a shelf slope regardless of location). Admittedly, neither the LME approach nor the shelf type approach is perfect, but we feel the LME method is most appropriate for our study. Our focus on different bathymetric depth intervals is meant to be a sort of middle ground, since it partially organizes our results by different bathymetric features within a given geographic region. Our decision to focus on LMEs and depth intervals is further motivated by their use in the management of marine resources around North America. For example, it is common for decision makers to consider what depth range benthic and demersal species might be found in when setting fishing guidance. By providing a breakdown of BMHW characteristics with depth interval, we provide a useful metric for this community to reference.

In response to these comments, we have added text to the Methods section to make these points.

Line 131 – I am assuming that the reporting of the R-values (i.e. no significance level) is following the approach of Jacox et al. (2020), but this should be stated.

- Thank you for catching this oversight. We have updated the Methods section to indicate that we are following the approach of Jacox et al. (2020).

Line 191- Fewing and Brown (FMS, 2019) would be another potential reference for this event

- We have added the suggested citation to this line.

Line 402- Is there a rationale for using the Jacox et al. (2020) over the Hobday et al. (2016)? What should be the guiding principle for this selection? I am guessing the motivation will be similar to that of Jacox et al. (2020) which should be briefly mentioned. Over the shelf where temporal and spatial scales are reduced relative to the open ocean one might argue that the 5-day methodology of Hobday et al. (2016) might be more appropriate. Maybe it doesn't make much of a difference? Regardless, this does raise the question about when/why to use different ways of defining marine heatwaves. This could be a discussion for a different paper though.

- As you alluded to, we chose to use monthly means for the reasons outlined in Jacox et al. (2020). Namely that: (1) As we'll show, the decorrelation timescale of the ocean can range from weeks to over a year, and thus we believe a minimum MHW duration of a month is more representative of the longer time scales of variability in the ocean; and (2) The most impactful MHWs in history have lasted at least one month (Hobday et al. 2018; Frölicher et al. 2018).

Regarding the time and spatial scale of bottom water temperature (BWT) variations, water parcels below the mixed layer are well-insulated from rapid temperature changes associated with atmospheric turbulent heat fluxes. As a result, it is reasonable to hypothesize that BWT variability might actually be lower frequency than the sea surface temperature (SST) variability at the same location. We test this hypothesis by calculating the decorrelation timescale of BWT and SST anomalies in each LME using GLORYS daily mean data from 1993-2019. We calculate the decorrelation timescale using the methods of DelSole (2001), which accurately represent the decay rate of temperature anomalies in the presence of oscillatory behavior (e.g., ENSO).

Examining Figure R3, we see that the decay rate of BWT anomalies is greater than 15 days for the majority of the LMEs, with the exception of the Mississippi River outflow region in the Gulf of Mexico and the outer portions of the Southeast US continental shelf. Indeed, the BWT decay rate is greater than 30 days for the vast majority of the LMEs, regardless of bottom depth (Table R2). Even for SST anomalies, which are subject to high-frequency turbulent heat fluxes, the decay rate is greater than 15 days for nearly all regions (Figure R4 and Table R2). Further, the BWT decorrelation timescale tends to greatly exceed the SST decorrelation timescale (Figure R5), consistent with our hypothesis that bottom waters tend to be insulated from higher frequency atmospheric forcing.

Based on Figure R3-R5 and Table R2, as well as the reasons stated in Jacox et al. (2020), we believe our use of monthly mean data is justified. To address this comment, we have included Figures R3-R5 in the Supplementary Information and added text to the Methods section to support our decision.

References:

- Delsole, T. (2001). Optimally persistent patterns in time-varying fields. *Journal of the Atmospheric Sciences*. [https://doi.org/10.1175/1520-0469\(2001\)058<1341:OPPITV>2.0.CO;2](https://doi.org/10.1175/1520-0469(2001)058<1341:OPPITV>2.0.CO;2)
- Frölicher, T. L., & Laufkötter, C. (2018). Emerging risks from marine heat waves. *Nature Communications*, 9(1), 650. <https://doi.org/10.1038/s41467-018-03163-6>
- Hobday, A. J., Spillman, C. M., Eveson, J. P., Hartog, J. R., Zhang, X., & Brodie, S. (2018). A framework for combining seasonal forecasts and climate projections to aid risk management for fisheries and aquaculture. *Frontiers in Marine Science*, 5(APR). <https://doi.org/10.3389/fmars.2018.00137>

Figure R3 Decorrelation timescale (days) of daily mean BWT anomalies from 1993-2019. Blue shading denotes decorrelation timescales of less than 15 days.

Figure R4 Same as Figure R3, but for daily mean SST anomalies.

Figure R5 Difference in decorrelation timescales between daily mean BWT and SST anomalies from 1993-2019.

Table R2 Decorrelation timescales for daily mean bottom water temperature (BWT) and sea surface temperature, averaged in different depth intervals. Values are bolded where the decorrelation timescale is less than 30 days.

Average Decorr. Timescale (days)	0-50m		50-100m		100-150m		150-200m		200-250m		250-300m		300-350m		350-400m	
	BWT	SST	BWT	SST	BWT	SST	BWT	SST	BWT	SST	BWT	SST	BWT	SST	BWT	SST
East Bering Sea	143	138	228	212	205	188	199	169	200	169	195	164	192	174	157	167
Gulf of Alaska	77	79	127	104	155	104	151	104	144	112	141	109	108	110	96	116
California Current	82	66	118	80	115	80	99	81	84	85	76	78	65	75	61	77
Gulf of California	58	54	96	54	117	54	133	55	126	57	176	59	154	56	201	58
Gulf of Mexico	38	26	54	30	45	30	42	28	42	28	45	28	49	27	51	28
Southeast US	25	22	31	21	30	22	25	20	26	20	19	18	20	18	22	18
Northeast US	53	36	83	54	119	53	200	53	215	54	150	53	99	53	69	47
Scotian Shelf	38	35	94	42	136	46	173	44	208	43	172	44	212	44	205	44
Labrador Shelf	34	34	81	57	139	38	139	38	154	42	183	47	185	52	185	51

Line 410 What is the reasoning of the 90th percentile from the 3 month distribution as opposed to just the monthly distribution? Seems like this could artificially bias the distribution of marine heatwave (e.g., peak coolest/warmest months will have lower values in the distributions used to calculate the 90th percentile?)

- The use of 3-month distributions was done in order to generate a more robust sample for calculating the seasonally evolving 90th percentile from the relatively short GLORYS record (1993-2019). Our results and conclusions are not sensitive to this choice. In response to this comment, we have added text to the Methods to make this point.

Figure 4 – Colorbar has two shades in 0-1 month range, but with monthly data, I am not sure how a marine heatwave could last less than 1 month (and I don't see any colors in the shades). Maybe change the color bar to remove the 0-1 range.

- You are correct, thank you for catching this oversight. We have adjusted the duration colorbars throughout the manuscript to more appropriately start at 1 rather than 0.

Reviewer #2 (Remarks to the Author):

Summary:

Using the GLORYS 1/12° model output (1993-2019) the authors have calculated marine heatwaves (MHW) at the bottom (BHW) and the surface (SHW) in the North American LME for grid pixels < 400 m deep in order to 1) document the intensity and duration of BHW and 2) compare them to SHW. As the body of literature on SHW is very well established, the authors are able to mainly focus on the BHW results. With SHW being useful here for comparison.

- Thank you for the time and effort you put into your review of our manuscript. You will find responses to your individual comments below. Please note that Line references in our responses are for the *revised* manuscript.

What are the noteworthy results?

The authors show some very interesting differences in the intensity and the spatial/temporal occurrence of BHW and SHW both within and across LME. Within LME by showing how there can be time lags between BHW and SHW, or even the occurrence of one without the other. Between LME by showing how the neighbouring LME can experience similar BHW and/or SHW, often with a time delay or other notable difference. This is attributed largely to the shallowness of a given pixel in that these are more likely to have a MLD that can extend to the bottom. The authors also provide an interesting discussion on areas where this relationship is less linear, likely due to areas with more bathymetric variability. An interesting area of research in the future to be sure.

Will the work be of significance to the field and related fields? How does it compare to the established literature? If the work is not original, please provide relevant references.

The work will certainly be of significance to the field of MHW studies and to related fields that study these events in physical oceanography and ecology. And likely many more (e.g. demersal fisheries). It compares very well to the established literature and is an obvious and much needed progression of it. There have been many other studies that look at BHW in some capacity, but none at this scale, and the authors reference these studies here.

Does the work support the conclusions and claims, or is additional evidence needed?

The work here definitely supports the conclusions/claims made, but I found the paper to be a bit light on them considering how impressive the results presented here are. Looking at Figure 8 in particular there are dozens of features whose investigations could require an entire fleet of researchers to follow up on, which would in turn be very interesting and important contributions to the literature. That being said, by not making any claims about the development of BHW over time the authors have avoided any complications from the one issue I have with the analysis.

- Thank you for your thoughtful comments on this subject. We agree that it would be worthwhile to provide more physical insight into the evolution of BMHW events in each of these LMEs, as well as a detailed study of the individual events identified in Figure 8. However, as you pointed out, doing so would require a dedicated focus which is beyond the scope of a single analysis. It is our hope that this paper will provide motivation for more region-specific studies aimed at quantifying the physical drivers of BMHWs in greater detail. To support this goal, we have provided a number of reasonable and readily testable

hypotheses on which future studies can build (see Lines 318-335). Further, we have already begun a separate analysis of the mechanism behind BMHWs in the California Current LME (CCS). Preliminary results indicate that the physical drivers of BMHWs in the CCS are multi-faceted and complex, with thermocline displacements driving BMHW events from Baja California to the Southern California Bight and upwelling variability producing BMHW events further north. We are further collaborating with researchers at GFDL on a third study to diagnose the mechanisms behind BMHWs in the Northeast US LME.

Are there any flaws in the data analysis, interpretation and conclusions? Do these prohibit publication or require revision?

An ongoing topic of debate in the field of MHW study is whether or not to first de-trend time series before detecting events, with the authors using here a well-established de-trending methodology. The argument between the two being (which I agree with) that de-trending better allows an investigator to isolate the historic/present drivers of extreme events independent of the long-term climate change signal. A paper however has recently been published (see specific comments below for reference/link) that quantifies the error that linear de-trending introduces into MHW results. Notably that it tends to over-inflate the intensity/duration of events at the beginning of time series because the rate of warming has generally been faster over the last decade. This also tends to cause a dip in the duration/intensity of events in the middle of the time series, as one may see in the occurrence of BHW in the E Bering Sea in Figure 5. I think that the analysis as it is now is fantastic and nothing should be changed, but that the manuscript would benefit from one or two sentences, either in the methods or discussion section, that addresses this potential issue. Another thought I had while reading this was that rates of warming between the bottom and the surface are likely to be different for much of study areas here. Meaning that de-trending the time series before detecting and comparing results will likely introduce different biases (of course, so would not de-trending the data) at the bottom and surface. I would be interested to read what the authors thought of that potential issue. As they see fit to do so.

- While we agree that the rate of warming has accelerated, we note that the non-linearity of global warming in the historical record is most apparent for long temperature timeseries that begin prior to the recent acceleration in warming (i.e., before about 1980). (Note that the statistical models presented by Wang et al. (2022) use timeseries spanning 100 years or more). Nevertheless, we have repeated our analysis by removing trends based on a 2nd-order polynomial fit and found that our results are not significantly different from those with a linear trend removed. Therefore, we feel it is reasonable to approximate the warming trends in this dataset as linear. Subsequently, while it is possible that the linear warming rates of the surface and bottom are different, removing their respective trends effectively normalizes the two fields, allowing them to vary on a more level playing field.

We further agree that the decision of whether to account for warming trends can certainly be an important factor when defining MHW events (either at the surface at the bottom). In this study, we chose to detrend the data because we are interested in representing the temperature variability throughout the record, independent of any long-term mean changes. Nevertheless, in order to test the sensitivity of our results to this methodological choice, we have repeated our analysis using the raw (i.e., not detrended) bottom temperature and sea surface temperature anomalies. For brevity, we present a subset of these results here

(Figures R6 and R7 and Tables R3 and R4). However, the following conclusions are consistent across our full suite of analyses.

Comparing Figure R6 and Figure R7 to main text Figure 5 and Figure 8, respectively, we see that the presence of a trend has very little effect on the spatial extent or average intensity of major BMHW/SMHW events in most of the LMEs. This is further supported by comparing the average BMHW intensity and duration for different depth intervals in each LME for raw versus detrended data (Tables R3 and R4). The clear exceptions are the Northeast US LME, the Scotian Shelf LME, and to a lesser extent the Labrador LME. These regions show greater spatial extent and SMHW/BMHW intensity from 2012-2019 for the raw data than for the detrended data (Figures R6 and R7), as well as larger average intensities and durations for nearly every depth interval (Tables R3 and R4). This is not surprising given that the Northwest Atlantic Ocean has some of the strongest warming trends on the planet (Pershing et al. 2015). Regardless of these slight differences, we do not believe our decision to detrend the data before analysis ultimately affects our primary conclusions that: (1) Marine heatwaves along the ocean bottom can have intensities and durations that exceed their surface counterparts; and (2) While SMHW and BMHW events are connected by MLD variations, BMHW events can occur without a clear surface signature.

In response to this comment, we have added a paragraph to the Methods to summarize these points. We have also added Figures R6-R7 and Tables R3-R4 to the Supplement along with additional text in the Methods to summarize their results.

References:

Pershing, A. J. et al. Slow adaptation in the face of rapid warming leads to collapse of the Gulf of Maine cod fishery. *Science* **350**, 809–812 (2015).

Figure R6 Same as Figure 5 of the main text, but for the raw (i.e., not detrended) bottom temperature data.

Figure R7 Same as Figure 8 of the main text, but for the raw (i.e., not detrended) sea surface and bottom temperature data.

Table R3 Average BMHW intensity (°C) for grid cells falling within different bottom depth intervals in each LME. Values are given for both raw and detrended bottom temperature data. Note that the detrended values are simply the gray dots in Figure 3.

Average Intensity (°C)	0-50m		50-100m		100-150m		150-200m		200-250m		250-300m		300-350m		350-400m	
	Raw	Det	Raw	Det	Raw	Det	Raw	Det	Raw	Det	Raw	Det	Raw	Det	Raw	Det
East Bering Sea	2.1	1.9	1.7	1.6	1.0	1.0	0.9	0.9	0.7	0.7	0.6	0.5	0.5	0.4	0.3	0.3
Gulf of Alaska	1.6	1.5	1.3	1.3	1.0	1.0	0.8	0.8	0.6	0.6	0.6	0.5	0.6	0.5	0.5	0.5
California Current	2.4	2.4	2.9	2.9	1.8	1.8	1.2	1.2	1.0	0.9	0.9	0.8	0.8	0.8	0.7	0.7
Gulf of California	2.0	2.0	3.1	3.1	2.6	2.6	1.9	1.9	1.2	1.2	1.0	0.9	0.8	0.7	0.8	0.7
Gulf of Mexico	1.6	1.5	1.6	1.5	1.7	1.6	1.8	1.7	1.7	1.6	1.6	1.4	1.5	1.5	1.5	1.4
Southeast US	1.7	1.7	1.9	1.8	2.2	2.1	2.3	2.2	1.9	1.7	1.7	1.6	1.4	1.3	1.2	1.1
Northeast US	2.3	2.2	2.4	2.2	1.9	1.7	1.6	1.3	1.6	1.3	1.7	1.4	1.7	1.3	1.5	1.3
Scotian Shelf	1.7	1.5	1.9	1.6	2.3	1.9	2.0	1.6	1.8	1.4	1.5	1.0	1.3	0.9	1.2	0.9
Labrador Shelf	1.5	1.5	1.4	1.3	1.2	1.2	1.3	1.2	1.3	1.2	1.1	1.0	0.9	0.8	0.9	0.8

Table R4 Average BMHW duration (months) for grid cells falling within different bottom depth intervals in each LME. Values are given for both raw and detrended bottom temperature data. Note that the detrended values are simply the gray dots in Figure S2.

Average Duration (months)	0-50m		50-100m		100-150m		150-200m		200-250m		250-300m		300-350m		350-400m	
	Raw	Det	Raw	Det	Raw	Det	Raw	Det	Raw	Det	Raw	Det	Raw	Det	Raw	Det
East Bering Sea	2.5	2.3	2.9	2.8	3.1	3.0	2.9	2.9	3.1	3.0	3.1	2.9	3.1	2.9	3.0	2.4
Gulf of Alaska	2.3	2.2	2.9	2.7	3.2	3.0	3.1	2.9	3.0	2.8	3.2	2.9	3.0	2.8	2.5	2.3
California Current	2.9	2.7	3.4	3.3	2.7	2.7	2.3	2.3	2.2	2.2	2.1	2.0	2.0	1.9	1.8	1.8
Gulf of California	1.8	1.7	2.3	2.3	2.9	2.8	2.8	2.8	2.3	2.4	2.2	2.3	2.2	2.3	2.6	3.9
Gulf of Mexico	1.6	1.5	1.8	1.7	1.6	1.6	1.5	1.5	1.6	1.4	1.6	1.5	1.5	1.6	1.6	1.6
Southeast US	1.4	1.4	1.5	1.4	1.6	1.4	1.5	1.4	1.6	1.4	1.5	1.4	1.4	1.3	1.4	1.4
Northeast US	1.9	1.9	2.0	1.8	2.2	2.0	2.9	2.5	3.5	2.9	2.7	2.1	1.9	1.7	1.8	1.6
Scotian Shelf	1.8	1.6	2.4	2.2	2.8	2.4	3.0	2.4	3.3	2.5	3.3	2.2	4.3	2.4	4.0	2.7
Labrador Shelf	1.7	1.7	2.2	2.2	2.5	2.4	2.7	2.5	2.9	2.6	3.1	2.6	3.0	2.4	3.5	2.6

Is the methodology sound? Does the work meet the expected standards in your field?

Yes, the methodology is well established and the work is at the highest level of quality in the field.

Is there enough detail provided in the methods for the work to be reproduced?

More or less, yes. It's not an issue of detail, but rather computing power, in reproducing the results. But that certainly is not the burden of the authors. Depending on the current publication requirements of Nature, I think that the MATLAB code used for the analysis should be uploaded to a public repository, rather than "Scripts are available upon requests."

- We have created a GitHub repository to store the MATLAB scripts used in our study. Codes can now be accessed at https://github.com/dillon-amaya/bottom_marine_heatwave. We have updated the Code Availability statement accordingly.

All the best,
-Robert Schlegel

Specific comments:

Abstract:

As I think that most non-physical oceanographers that read this will assume that these results are based on time series with the climate change signal not removed (i.e. non-de-trended), please somewhere point out to the reader that these results are based on de-trended time series.

- Thank you for the suggestion, but we'd prefer to leave methodological details such as this for the main text, especially given the strict word count for the abstract.

Intro:

Lns 59-67: As the authors prefer, consider adding Darmaraki et al. (2019) to this list of BHW analyses to include the Mediterranean. I am not a co-author.

- Thank you for bringing this paper to our attention. We have added it to the introduction at the suggested location.

Ln 73: "BTAs" here and throughout; I've noticed a trend in the heatwave literature where authors have begun no longer including the "s" after the plural form of acronyms. I think it looks a bit cleaner to remove the "s". Completely at the authors /editors discretion.

- Thank you for the suggestion, but we prefer to keep the "s". We have, however, updated the text throughout to limit the use of this stylistic choice.

Ln 93: "high-resolution"; I appreciate that this section is meant to be as direct as possible, but I think many readers would like to see the resolution of the model included here in parentheses. As the editing process prefers.

- We have added the resolution in parenthesis at the suggested location.

Results:

Ln 109: “NEUS” here and throughout; should this be “NE US”? It took me a moment to realise where this was as I initially thought it was an acronym for some climate index or other physical force. Same for “SEUS” etc.

- NEUS and SEUS are common acronyms used in the literature to refer to these LMEs. Therefore, we’d prefer to continue with their use here. However, to improve clarity, we have added the acronyms to the titles of the corresponding subpanels in each Figure.

Ln 114-123: Very interesting insight that I hadn’t realised just from looking at the figures beforehand.

- Thank you, we’re glad the description of Figures 1 and 2 is helpful.

Ln 126: “scatter diagrams” → “scatter plots”

- We have made the suggested edit.

Ln 144-152: Also very interesting insight.

- Thank you!

Ln 153: “varies strongly within and among” This phrase implies a statistical test (e.g. two way ANOVA) was performed.

- We have changed the wording of this line to read “The average duration of BMHW events also exhibits strong spatial variations across the different LMEs (Figure 4)”. We feel this more accurately introduces this figure without spuriously implying that a statistical test was performed.

Ln 166-168: Creating a spatial average of an LME does not show whether or not events co-occurring in time within it are spatially cohesive. Meaning that one can have many small patches of events caused by different forces. Though this is more of a surface signature, and I doubt that it is occurring at depth.

- This is a fair point. While our analysis in Figure 5 does not represent a “spatial average”, we acknowledge that it also does not show the extent to which nearby grid cells aggregate to produce cohesive events. However, given the spatial scale of each LME’s area (1000s km²), it’s unlikely that instances of large spatial extent (for example, months with >50% of the domain in BMHW conditions) would be the result of several small BMHW events that happen to co-occur independently of one another. That said we agree that our analysis cannot explicitly rule out that possibility.

In response to this comment, we have adjusted this Line to read: “In order to diagnose the prevalence of BMHWs on the broader LME-scale, we assess the spatial extent of these

events with time (Figure 5).” We have further adjusted the text to more appropriately refer to the events described in Figure 5 as “widespread” as opposed to “spatially cohesive”.

Lns 179-181: Indeed, when one linearly detrends a time series in a region highly influenced by ENSO, it is the Nino3.4 index that best correlates to the heatwave statistics.

- We note that these features are also present when calculating Figure 5 without detrending (Figure R6).

Lns 201-204: While it may be outside of the scope of this article, one could certainly devise a methodology that uses the results here in which one could conclude whether the events are spatially consistent or not. Perhaps an area of future research. I think it would be interesting in the LME with greater geomorphological variance to show the spatial differences.

- See our response to your similar comment above. We agree that it may be possible to identify individual, cohesive events, but we also agree that that sort of analysis is beyond the scope of this paper.

Ln 211: “Chen et al. 2021” → “Chen et al., 2021”

- We have made the suggested edit.

Lns 216-225: The north-south difference is interesting.

- Thank you, we’re glad you think so!

Lns 226-227: That’s surprising that spatial surface intensity is less variable than bottom. Or does this mean that SHW in a given LME tend to be the same event, so have a similar signal, whereas BHW can be different events, therefore having more variance?

- We hypothesize that the SMHW intensity maps are smoother because BMHW events are more likely to be bathymetrically constrained to certain regions within an LME, which is something we discuss at Lines 306-317. Whether bathymetrically isolated BMHW events should be considered “different events” is something we hope to investigate in future analyses focused on specific LMEs.

Lns 227 and 229: Switch the references to Figure S3 and S4.

- Thanks for catching this error. We have switched the order of these Figures in the Supplement so that these references in the main text are now correct.

Lns 227-229: This is difficult to understand.

- We agree and have removed this sentence.

Lns 285-287: Good to know.

- We're glad this description was helpful.

Lns 294-297: I think it would be good to provide a condensed form of this explanation to the caption for Figure 10. Having looked at the figures before reading the manuscript, it took me a moment to be certain which direction the X-axis was going.

- We have added the following example to the Figure 10 caption to make this clearer. "For example, grid cells in the upper right quadrant represent regions within each LME with high BMHW/SMHW synchrony and a deep MLD relative to the ocean bottom."

Summary and Discussion:

Ln 301: "high-resolution (1/12°)" Very nice. I've considered using this product before. But the size of the data... jikes.

- It is definitely a cumbersome dataset, but quite useful for regional analyses such as this. Please feel free to reach out to us if you'd ever like a subset of the dataset. We have it downloaded in its entirety on our servers. You may also be interested in the 0.25° product found here.
https://resources.marine.copernicus.eu/product-detail/GLOBAL_REANALYSIS_PHY_001_031/INFORMATION

Lns 327-330: I'm not convinced, given the crude nature of the spatial aggregation, that the authors can make this concluding statement.

- We feel this conclusion is justified given our assessment of Figures 1-3 and Figures S1-S3. These Figures clearly show strong regional variations in BMHW intensity and duration that align with different bathymetric features unique to each LME (e.g., the Gulf of Maine and the Mid-Atlantic Bight in the NEUS LME). In particular, the fact that different bathymetric features "light up" in maps of BMHW average intensity and not for SMHW average intensity (comparing Figures 2 and S3) is a further indication that bottom topography is critical in shaping the patterns described in this paper.

To address this comment, we have added text supporting these conclusions to Lines 306-317.

Lns 330-333: Certainly an interesting future avenue of study.

- We agree and look forward to delving into the LME-specific physical mechanisms in the near future. In fact, we have already begun investigating the physical drivers of BMHW in the California Current LME and we have ongoing collaborations with researchers in the Northeast US who are conducting a similar analysis of NEUS LME BMHW events.

Lns 334-350: The authors may want to remove some of this corresponding text from the results section as much of this has already been stated there.

- We have removed the redundant text from the Results section.

Lns 368-371: This would be amazing to have.

- We agree 100%!

Methods:

Lns 404-405: While for the last few years the disagreement about de-trending vs not de-trending time series before detecting MHW has been a philosophical one (based largely on whether or not the field of research in question is physical oceanography or ecology), there is now a publication that quantifies the error introduced into MHW results from linear de-trending time series. Please see Wang et al. (2022: https://journals.ametsoc.org/view/journals/atot/aop/JTECH-D-21-0142.1/JTECH-D-21-0142.1.xml?tab_body=pdf) for the specific statistical models used etc. That being said, the methodology used here is indeed well established, so I am not suggesting that anything be recalculated. Rather, I think that the authors should include one or two sentences explaining to the reader how the method of linear de-trending inflates the intensity of MHW earlier in a time series when linearly de-trended.

- Thank you for bringing this useful paper to our attention. As mentioned above, in response to this and other reviewer comments, we have added a paragraph to the Methods to discuss the issue of detrending. We have also added Figures R6-R7 and Tables R3-R4 to the Supplementary Information.

Code availability:

“Scripts are available upon requests.” Does not mean that the code is open access. It is best practice to list the GitHub (or otherwise) repo where the code is hosted.

- We have added the relevant MATLAB scripts to a GitHub repository and have added the relevant links to the Code availability section of the paper.

Figures:

Figure 1: A good start to the figures and useful to have.

- Thanks!

Figure 2: Also useful, but a bit unfortunate about the California Current panel having so much white space. No recommended changes though.

- We agree that it's unfortunate the California Current panel is mostly white space. It does, however, illustrate how narrow the continental shelf along the US west coast is when compared to other regions.

Figure 3: I like these 2-D histograms a lot. The grey dots are a useful addition.

- Thanks!

Figure 4: If the methodology analyses monthly data, how can the average duration of an event last for 0 or 0.5 months? Shouldn't then this (and other duration colourbars throughout) start at 1, rather than 0?

- You are correct, thank you for catching this oversight. We have adjusted the duration colorbars throughout the manuscript to more appropriately start at 1 rather than 0.

Figure 5: These results are mesmerising.

- Thank you!

Figure 6: The thin shelf edge / Gulf Stream signal in panel g for intensity differences is very interesting.

Figure 7: And how it's not present here for the duration.

- We agree with these two comments that this thin shelf feature is quite interesting. We plan to explore this and similar features with our collaborators in the Northeast US.

Figure 8: Overlaying the BHW and SHW results here is incredible. It is of course outside the scope of this paper, but an interactive website to explore these results would be awesome.

- This is certainly an interesting idea moving forward. We'll keep it in mind, thank you!

Figure 9: I find this to be the most interesting of all of the map-based figures. It does a great job of showing how different the bottom-surface relationship is over The Grand Banks than almost all LME.

- Thank you!

Figure 10: I'm not completely sure what the shading in this figure is showing. Perhaps explain it again in the caption in more detail.

- The shading is as in other similar histogram figures, but for the probability of a grid cell falling within a given synchrony and MLD/bathymetry ratio interval. In response to this and other reviewer comments, we have added text to the Figure 10 caption to further clarify its meaning.

Figure S1: Future work could cut the LME into smaller areas to see which of the upward curves of dots that emerge from the cloud (as in panel c) relate to which smaller regions.

- We totally agree! In fact, this is something that we are actively collaborating on with our colleagues in the Northeast US. Our aim is to better understand the regional mechanisms that drive BMHW in the NEUS LME.

Figure S2: Interesting.

- Thanks!

Figure S3: A lot of interesting relationships to unpack here in the future.

- Agreed, we are hoping to delve deeper into each LME with subsequent analyses, starting with the California Current LME and NEUS LME.

Figure S4: Helpful for understanding the referencing statement in the manuscript.

- Thanks!

Figure S5: This is another very interesting way of comparing surface and bottom temperature anomalies.

- We agree! It's amazing how often BMHW are warmer than SMHWs.

Figure S6: I find the two different statistical comparisons here very useful. Why not add these to the figures in the main manuscript?

- Unfortunately, our submission is limited by Nature Communications to 10 display items. While Figures S6 and S7 (now Figures S9 and S10) are indeed useful, we do not feel they should replace any of the existing Figures in the main text.

Figure S7: This does a good job of explaining the patterns in Figure 9. Perhaps reference this figure in that caption.

- While Figure S7 (now Figure S10) is informative, it does not directly relate to the calculation of Figure 9. It does, however, contribute to the histograms presented in Figure 10. We have added a reference to Figure S10 in the Figure 10 caption.

REVIEWERS' COMMENTS

Reviewer #1 (Remarks to the Author):

The authors have defended their positions well and incorporated revisions where reasonable. I think the authors should be commended for their efforts to validate the GLORYS data set. My concerns have been addressed.

Reviewer #2 (Remarks to the Author):

I think that the authors have provided a more than thorough response to all of my comments. I am happy to recommend this article for publication in its current form.